# Consistent predator-prey biomass scaling in complex food webs

**Daniel M. Perkins** [1] ✉, **Ian A. Hatton**[2] ✉, **Benoit Gauzens** [3,4],
**Andrew D. Barnes** [5], **David Ott**[6], **Benjamin Rosenbaum** [3,4],
**Catarina Vinagre**[7,8] **& Ulrich Brose** [3,4]

The ratio of predator-to-prey biomass is a key element of trophic structure that is typically investigated from a food chain perspective, ignoring channels of energy transfer (e.g. omnivory) that may govern community structure. Here, we address this shortcoming by characterising the biomass structure of 141 freshwater, marine and terrestrial food webs, spanning a broad gradient in community biomass. We test whether sub-linear scaling between predator and prey biomass (a potential signal of density-dependent processes) emerges within ecosystem types and across levels of biological organisation. We find a consistent, sub-linear scaling pattern whereby predator biomass scales with the total biomass of their prey with a near ¾-power exponent within food webs - i.e. more prey biomass supports proportionally less predator biomass. Across food webs, a similar sub-linear scaling pattern emerges between total predator biomass and the combined biomass of all prey within a food web. These general patterns in trophic structure are compatible with a systematic form of density dependence that holds among complex feeding interactions across levels of organization, irrespective of ecosystem type.

Understanding the processes that drive the structure and functioning of ecosystems is a fundamental goal in ecology. The ratio of predator-to-prey biomass provides a key measure of trophic structure and community dynamics[1–3] and is linked to many ecosystem functions and services[4,5]. When partitioning individuals or species into trophic levels, the distribution of biomass along food chains tends to form a characteristic 'pyramid' pattern with greater standing stocks of biomass at lower trophic levels[1]. That is, biomass pyramids tend to be 'bottom heavy' in size-structured assemblages, where trophic level increases with body size[3], although this pattern is by no means universal[6]. Theoretical syntheses have highlighted a plethora of possible mechanisms that can drive energy flow through food webs and thus differences in the shape of biomass pyramids and the ratio of predator-to-prey

biomass[6,7]. However, the principal mechanisms responsible for driving these patterns in natural systems remains uncertain because of a lack of empirical data, and investigations of how these patterns may change along environmental gradients are still in their infancy.

A previous finding highlights remarkable regularity in how the ratio of predator-to-prey biomass changes across a gradient of prey biomass in both aquatic and terrestrial systems[8]. Predator biomass, $y$, (e.g. total biomass of lions, hyenas, and other large carnivores) was found to scale with the biomass of their prey, $x$, (e.g. dik-dik, buffalo and other herbivores) in a sub-linear fashion on double logarithmic scales[8]. This 'predator-prey power law'[8], therefore, takes the form: $y = cx^k$, where $c$ is the normalisation coefficient and $k$ is the dimensionless scaling exponent. The power-law exponent, $k$, has been found

[1]School of Life and Health Sciences, Whitelands College, University of Roehampton, London SW15 4JD, UK. [2]Max Planck Institute for Mathematics in the Sciences, Leipzig 04103, Germany. [3]EcoNetLab, German Centre for Integrative Biodiversity Research (iDiv) Halle-Jena-Leipzig, Leipzig, Germany. [4]Institute of Biodiversity, Friedrich Schiller University Jena, Jena, Germany. [5]Te Aka Mātuatua - School of Science, University of Waikato, Private Bag 3105, Hamilton, New Zealand. [6]Centre for Biodiversity Monitoring (Zbm), Zoological Research Museum Alexander Koenig, Adenauerallee 160, 53113 Bonn, Germany. [7]Marine and Environmental Sciences Centre, Faculdade de Ciências da Universidade de Lisboa, Lisbon, Portugal. [8]Centre of Marine Sciences, University of Algarve, Faro, Portugal. ✉ e-mail: daniel.perkins@roehampton.ac.uk; i.a.hatton@gmail.com

to be <1 and consistently close to ¾ across ecosystem types, implying biomass pyramids become systematically more bottom-heavy with increasing prey biomass (Fig. 1a). These empirical patterns could be underpinned by systematic changes in total prey production available to predators[8,9] – that is, because predator biomass and prey productivity are linearly related[8,10]; if predator biomass is sub-linearly related to prey biomass (Fig. 1a), then prey productivity should also be sub-linearly related to prey biomass. Two possible reasons explain the sub-linear scaling of prey productivity with prey biomass: (1) constraints on individual rates which scale allometrically with body mass[11] that could lead to sub-linear community scaling with a systematic relation of body size to biomass[12], and/or (2) density-dependent effects - i.e. submaximal individual growth[8]. Hatton et al.[8] found that mean prey body mass varied little over the prey biomass gradient, and thus their results are consistent with the role of density-dependent processes driving sub-linear predator-prey biomass scaling.

Natural communities are, however, rarely neatly conceptualized by food chains formed by groups of organisms assigned to discrete trophic levels such as carnivores or herbivores[13,14]. Most prey are shared by many predators and many taxa are omnivorous (feeding at more than one trophic level)[14]. Together this gives rise to a complex web of interactions where a consumer's trophic height can be a non-integer value (Fig. 1b), and where energy is channelled via diverse pathways. Whether a general predator-prey power law applies in complex food webs is therefore uncertain as various processes related to food web complexity could alter this relationship. For instance, food chain theory suggests omnivores are able to route energy up food webs more efficiently since they are capable of drawing energy directly from various resource pools and can side-step constraints imposed by consuming taxa at intermediate trophic levels that act as 'energetic middlemen'[6]. Size spectrum theory, on the other hand, suggests higher biomass can also potentially be sustained for large predators that feed

'down' the food web[3,15] - i.e. on the lowest trophic levels, which are typically the smallest organisms - because mass-specific production (and the production-to-biomass ratio) is greater for smaller organisms[16].

Here we develop a framework for quantifying predator-prey biomass scaling in complex food webs (Fig. 1) and apply this method to 30 freshwater (stream), 66 marine (intertidal rock pool) and 45 terrestrial (forest soil) food webs (Table S1), spanning a broad gradient in community biomass. We calculate the biomass of each taxon in a web and the total biomass of all prey for each predator, accounting for prey items shared by multiple predators (see Methods). We also test how the variation in the degree of omnivory and predator-prey body mass ratios (PPmR) could impact biomass scaling or residual variation in the trophic biomass relation. This compilation of biomass data across trophic levels, ecosystems and ecosystem types, offers the opportunity to test the ubiquity of the predator-prey power-law exponent in a diverse set of omnivorous food webs, with the potential to offer new insight into energy flow. Specifically, we ask: (i) how does predator and prey biomass scale within (Fig. 1c) and across (Fig. 1d) complex food webs, and does a general power-law hold across levels of biological organization; (ii) what role does omnivory and predator-prey body mass ratios have on the scaling pattern and (iii) is the association between predator and prey biomass underpinned by changes in prey density or the average size of prey? Our results reveal fundamental similarities in predator and prey biomass scaling within and across diverse food webs, providing a basis to link biomass distributions across levels of biological organization (Appendix S1).

## Results
### Predator-prey biomass scaling within webs
To characterise within-web scaling (Fig. 1c), we constructed relationships between ($\log_{10}$) predator biomass and ($\log_{10}$) total biomass of their prey for each web (Figs. S1–S3). Linear mixed-effects models were used to determine the scaling exponent ($k$) of the power-law for each ecosystem type in a single pass (Methods). Doing so revealed that the average exponents were all sub-linear and similar among ecosystem types, although confidence intervals were wide: freshwater $k = 0.61$ (95% CI 0.50 to 0.71), marine $k = 0.74$ (CI 0.66 to 0.82), and terrestrial $k = 0.75$ (CI 0.66 to 0.83) ecosystems, respectively (Fig. 2; Table S2). Moreover, the improvement in model fit going from the most complex model, which assumes a different power-law exponent for each ecosystem type, to a null model, which assumes a common exponent for all ecosystem types ($\bar{k}$, see Methods), provided only weak evidence that ecosystem type affects the scaling exponent (likelihood ratio test: $\chi^2 = 4.83$, d.f. = 2, $P = 0.0894$; Table S3). Thus, predator-prey biomass scaling within ecosystem types (freshwater streams, marine rock pools and terrestrial soils) can be characterized by the same average power-law exponent: $\bar{k} = 0.71$ (95% confidence interval of 0.66–0.76; Fig. 2). This near ¾-power, sub-linear scaling regime signifies that, as prey biomass doubles (an increase of 100%), predator biomass only increases by approximately 64%. However, as indicated by the significance of the random-effects terms in the model (Table S3), there was notable variation in exponents between food webs – i.e. at local scales.

To investigate the role of predator-prey body mass ratios (PPmR) and omnivory on the predator-prey biomass relation, we analysed these variables as additional terms in the mixed-effects model (see Methods). We define predator omnivory to be the variance in the trophic levels of its prey species (Methods). Inclusion of these fixed effects increased the proportion of explained variation ($R^2$) in predator biomass from 0.53 (Table S3) to 0.67 (Table S4). Moreover, this analysis revealed that residual predator biomass increased significantly with PPmR in each of the three ecosystem types (Fig. 3a–c). The slope of these relationships differed between ecosystem types as indicated by the significant 'PPmR × Ecosystem type' interaction term in the

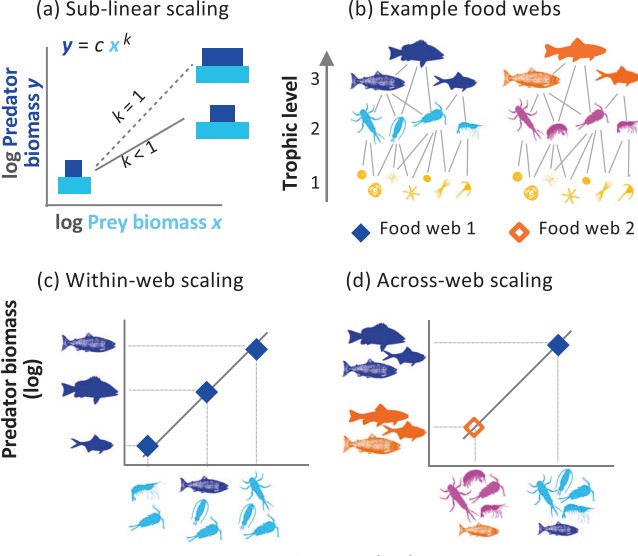

**Fig. 1 | Schematic of predator-prey biomass scaling across levels of organisation. a** The predator-prey power law exponent, $k$, describes relative changes in pyramid shape along a prey biomass gradient, with $k = 1$ denoting no relative change. **b** Trophic interactions in nature give rise to a complex web of interactions where a consumer's trophic height can take on a non-integer value. **c** Within-web scaling relations are such that each data point represents the biomass of different predator taxa plotted against the total biomass of their prey, within a single food web (see also Fig. 2). **d** Across-web scaling relations represent the total biomass of all predators plotted against the total biomass of all prey for each distinct food web (see also Fig. 4). In all cases, the abundance and mean body mass of all species are used to calculate trophic biomass.

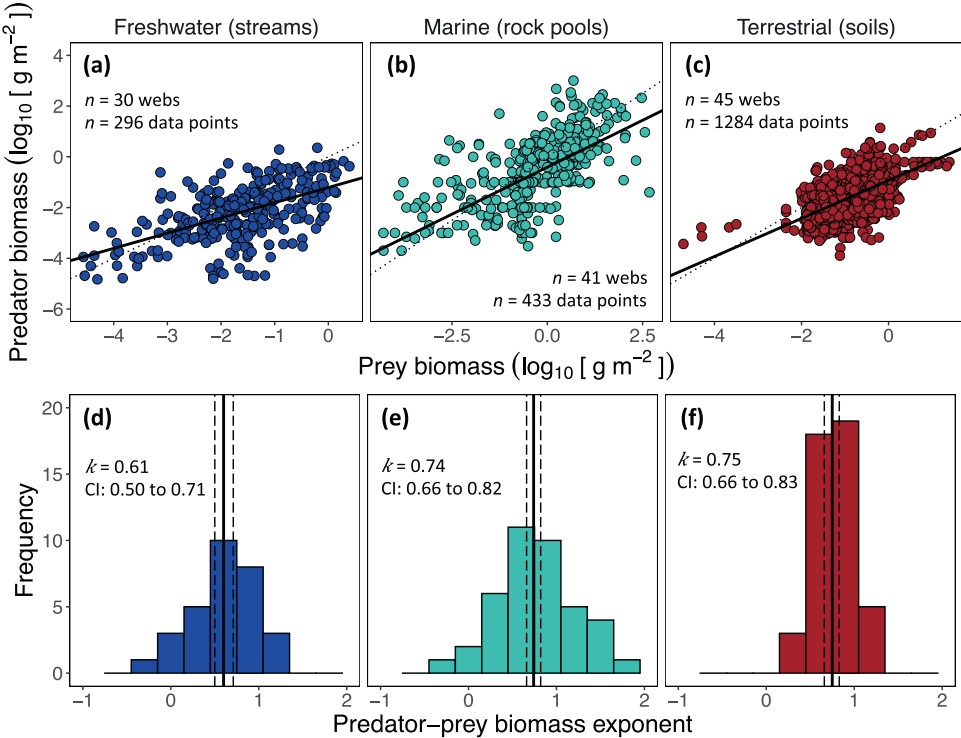

**Fig. 2 | Within-web predator-prey biomass scaling among ecosystem types.**
**a–c** Each data point is a different predator node and the total biomass of its prey within a food web (dashed line represents the 1:1 line). The fitted lines represent the mean power-law exponents for each ecosystem type: **a** $n = 30$ food webs; **b** $n = 41$ food webs; **c** $n = 45$ food webs. (Separate plots for each food web are shown in Figs. S1–S3). **d–f** The distribution of power-law exponents among food webs yields an average (solid line) that is statistically indistinguishable between ecosystem types and is close to ¾. Power law exponents and 95% confidence intervals (dashed lines) were determined from linear-mixed effects modelling (Table S3).

model (LME: F = 45.3, d.f. = 2, $P < 0.0001$; Table S4) with predator biomass increasing with PPmR more strongly in marine webs (Fig. 3a, b). A significant 'Omnivory × Ecosystem type' interaction was also evident (LME: F = 73.54, d.f. = 2, $P < 0.0001$; Table S4) with predator biomass residuals increasing with the extent of predator omnivory in soil webs, but only a weak association was obtained in marine webs and there was no evidence of a significant relationship in the freshwater webs (Fig. 3d, e; Table S2).

**Predator-prey biomass scaling across webs**
To investigate if similar predator-prey biomass scaling patterns emerge across webs, and thus whether within-web patterns can be scaled-up to whole ecosystems (Appendix S1), we summed the biomass of all predators and all prey within each food web. The across-web analysis (see Methods) revealed sub-linear and remarkably similar power-law exponents for each ecosystem type: freshwater $k = 0.66$ (95% CI: 0.43–0.90), marine $k = 0.65$ (CI: 0.47–0.83), and terrestrial $k = 0.67$ (CI: 0.26–1.09) ecosystems, respectively (Fig. 4; Table S2). There was no evidence that ecosystem type affected power-law exponents (F-test comparing a null model with a single power-law to an alternative model with separate power-law for each ecosystem type: ANCOVA: $F_{2,135} = 0.05$, $P = 0.9897$). The average exponent among ecosystem types, $\bar{k}$, was 0.66, similar to that observed for within-web scaling of predator and prey biomass ($\bar{k} = 0.71$), with the 95% confidence intervals of 0.52 to 0.79 overlapping with those derived for within-web scaling (0.66–0.76).

We investigated the potential role of changes in prey body mass versus prey density in driving the observed across-web predator-prey biomass scaling. For biomass scaling to be the direct result of changes in prey body mass (and thus related to body mass allometry), mean prey body mass would be expected to scale proportionately with prey biomass ($k = -1$; Appendix S2). Within each ecosystem type, mean prey

body mass was unrelated to prey biomass (i.e. an exponent near 0; Fig. 5; Table S2), and there was no evidence that changes in mean prey size with prey biomass differed between ecosystem types (ANCOVA: $F_{2,131} = 1.40$, $P = 0.8692$). Thus, the consistent changes in predator biomass with prey biomass we observed were primarily associated with changes in prey densities rather than the average size of prey.

## Discussion
Here we provide a unified analysis of predator-prey biomass scaling in complex food webs. Doing so reveals a consistent sub-linear scaling pattern across levels of organization - from populations within webs to whole ecosystems – for freshwater, marine and terrestrial systems. This regularity in sub-linear predator-prey scaling among complex food webs from diverse ecosystem types has important implications for understanding energy flows in natural systems across large spatial gradients.

Within food webs, predator-prey biomass scaling was characterised by a near three-quarter power scaling relationship ($\bar{k} = 0.71$ across ecosystem types), revealing an approximately three-fold increase in predator biomass for every five-fold increase in prey biomass. When summing all predator and prey biomasses within a food web (Fig. 4), predator-prey scaling across webs followed a similar sub-linear scaling regime, with $k$ ranging from 0.65 to 0.67 between ecosystem types. That is, biomass pyramids became systematically more bottom-heavy as pyramid size increased along a biomass gradient (Fig. 1a). These ecosystem-level patterns are quantitatively consistent with previous analysis of predator-prey biomass scaling among distinct trophic groups, which also found sub-linear scaling with $k$ values between 0.66 to about 0.76[8,17,18]. The approach we introduce here permits expanding these analyses to more complex omnivorous feeding relations both among populations within webs and across webs in diverse ecosystems. The similarity in the scaling exponents

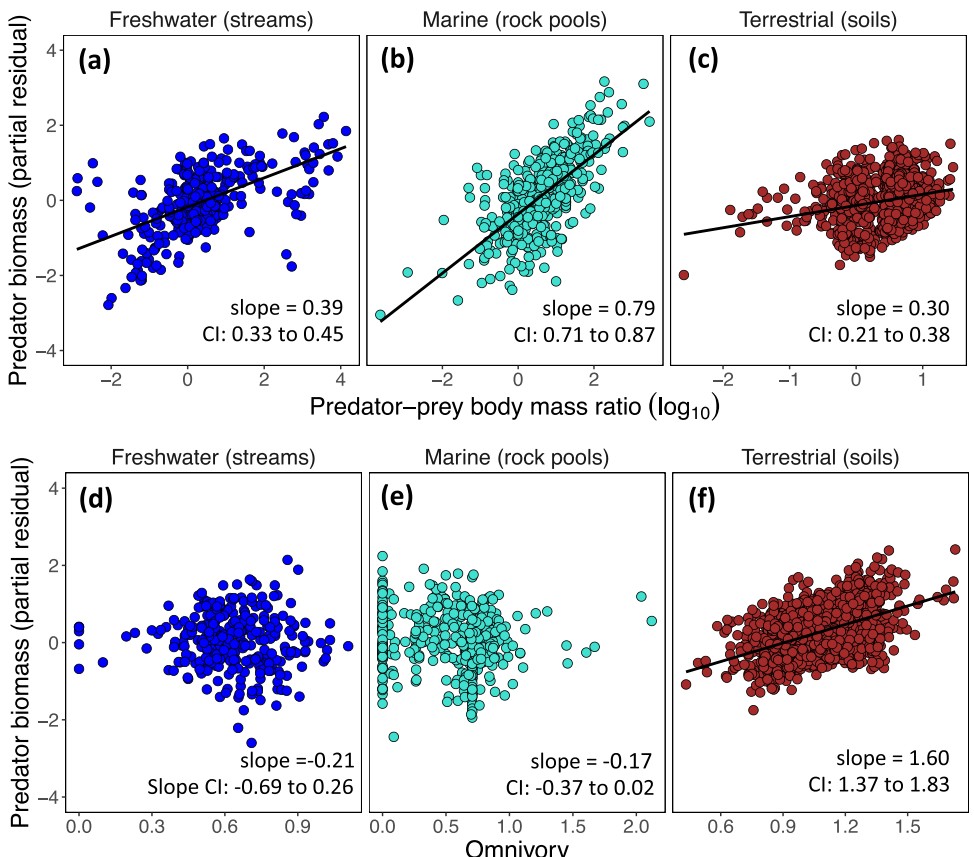

**Fig. 3 | Variables explaining residual variance in predator biomass among ecosystem types.** Data points are the partial residuals from the linear mixed-effects analysis of within-web scaling of predator biomass with prey biomass (Table S4). **a–c** relationships between residual predator biomass and log$_{10}$-transformed predator-prey body mass ratio (PPmR). **d–f** relationships between residual predator biomass and predator omnivory. Predator biomass deviations increase significantly with PPmR in all ecosystem types (**a–c**) and increase with predator omnivory in soil (**f**), but not freshwater or marine (**d**, **e**) food webs (regression lines not fitted).

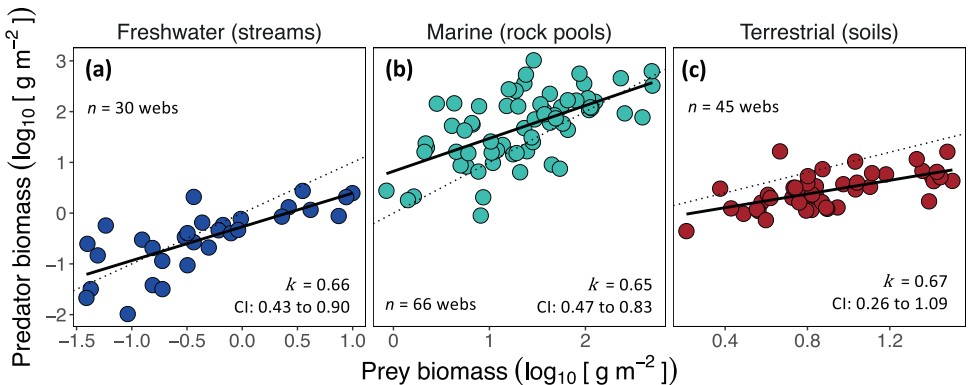

**Fig. 4 | Across-web predator-prey biomass scaling among ecosystem types.** Each data point is characterized by the total biomass of all predators and the total biomass of all prey in a food web (**a**) $n$ = 30 food webs; (**b**) $n$ = 66 food webs; (**c**) $n$ = 45 food webs. Power-law exponents and 95% CI were determined from analysis of covariance and are similar to those observed for within-web predator-prey biomass scaling (Fig. 2).

(and overlap in confidence intervals) of within- and across-web scaling suggest the existence of a general sub-linear scaling pattern, possibly signifying that fundamental constraints apply across levels of biological organization.

These results beg the question: where do these sub-linear scaling patterns originate? We are not aware of any ecological theory that is sufficiently general to encompass the diversity of community types in which sub-linear biomass scaling is observed (Appendix S2). Size spectrum theory, which aims to explain the observation that, for whole

ecosystems, biomass is approximately evenly distributed across logarithmic body size classes[19,20] would appear to be particularity relevant. However, static size spectrum models typically assume that the predator-prey body mass ratio (PPmR) and trophic transfer efficiency (ratio of predator to prey production), whilst inherently variable[21,22], do not vary systematically with prey biomass[19,23]. These measures indicate from which size class energy is obtained relative to predator body mass, and how efficiently that energy is utilized by any given predator to maintain its biomass. While these variables are thought to drive size

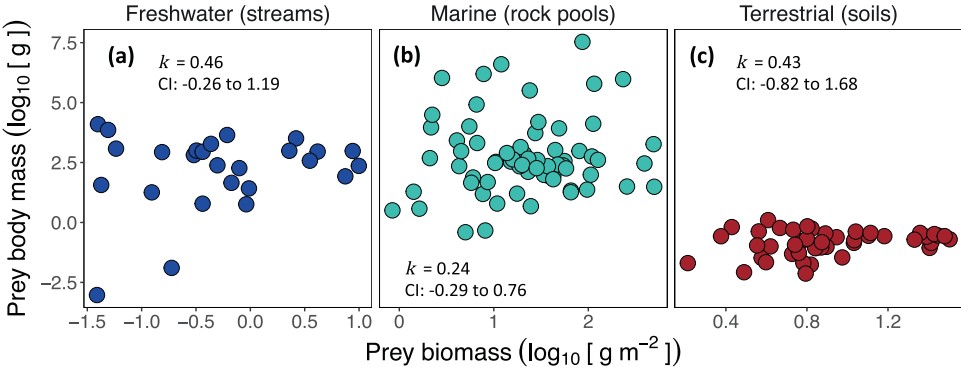

**Fig. 5 | Relationships between prey size and prey biomass. a–c** mean body mass versus prey biomass averaged for all prey in a food web (across-web scaling). The slopes, *k*, determined from analysis of covariance are all non-significant (all *p* > 0.05; Table S2) signifying that changes in prey biomass are primarily driven by changes in prey density rather than average prey size.

spectra scaling[3], they do not appear to be consistent with predator-prey biomass scaling observed in natural communities. Assuming both an even distribution of biomass across size classes, and a constant PPmR or transfer efficiency across a prey biomass gradient suggests an unchanging trophic biomass pyramid (all else being equal; Appendix S2), Therefore it is not clear how current size-spectrum models might account for sub-linear predator-prey biomass scaling.

Predator-prey theory, on the other hand, which models the dynamics of feeding interactions, has traditionally focused on two distinct trophic levels, rather than on networks of highly omnivorous food webs[24]. Equilibrium predictions from a range of simple predator-prey models are also not consistent with sub-linear predator-prey scaling without additional and likely questionable assumptions (Appendix S2). Although predator-prey theory can be made to accord with our observed patterns, it requires tuning the scaling of prey growth or other terms of the model. Furthermore, questions remain about how best to simulate a biomass gradient as well as how models should be generalized to multi-trophic food webs (Appendix S2).

Despite the lack of any general mechanism, it is reasonable to assume that predator biomass, at steady state, is maintained in proportion to prey production[8,10]. This would suggest that as prey biomass increases, their total production should scale near ~¾ to match the predator biomass they support. Density-dependent processes, such as competition for resources and other negative interactions among prey species, could thus cause *per capita* growth to decline sub-exponentially. We observed that changes in prey biomass were primarily driven by changes in prey density, rather than average prey body size, consistent with density dependent effects driving the sub-linear nature of predator-prey biomass relations, rather than allometric body mass effects. Clearly, however, ecological theory has further work yet to knit together the various patterns and processes to explain the consistency and generality of predator-prey scaling patterns.

Addressing predator-prey biomass scaling from a food web perspective allowed us to assess which node properties were associated with greater predator-prey biomass ratios. Our results go beyond prior theoretical studies[6,7] to provide empirical evidence that populations of highly omnivorous predators, as well as predator populations that feed down the food web on smaller, more productive, prey (i.e. a high predator-to-prey body mass ratio), tend to attain higher biomass stocks than predicted by their prey biomass alone. Interestingly, the role of these variables in driving predator biomass deviations appear to vary between ecosystem types: predator biomass increases more strongly with PPmR in rock pool webs, whereas predator omnivory only proved to correlate with predator biomass residuals in soil webs (Fig. 3). Further research would be instructive to understand if these are general patterns across different types of terrestrial and aquatic ecosystems. For instance, whilst rock pool webs display very similar

network topology and PPmR scaling as open marine webs[25,26], we might expect different scaling patterns in pelagic marine webs where trophic interactions take place in three dimensions[21], where ontogenetic diet shifts are common[27], and where food chains are long[13]. Adapting our food-web approach to quantify biomass scaling among size classes would likely be informative for tackling these additional complexities. Whilst predator biomass was associated with PPmR and omnivory (in soil webs), the consistent sub-linear predator-prey scaling regime within ecosystem types and across levels of organization, suggests that density dependent population growth might be the overriding driver of predator-prey biomass scaling.

The regularity in predator-prey scaling we observed could provide insight into baselines for the biomass structure of natural communities, which could be informative for assessing the effects of environmental impacts within ecological communities and ecological status. For instance within webs, deviations away from these baselines in the form of smaller power-law exponents (shallower slopes) could reflect local perturbations (e.g. acidification, warming, over-exploitation) which have a disproportionate impact among larger organisms at higher trophic levels[28]. Predator-prey biomass scaling could therefore offer a complementary approach to body size distributions and size spectra for evaluating ecosystem health[29]. A similar approach could be applied for scaling relations within species, where the same species occur in multiple webs. Doing so could reveal how the biomass of a given predator species responds to variation in prey availability. For instance, among the stream food webs studied here, two common fish species displayed the characteristic near ¾-power scaling pattern, whilst the biomass of salmonids, and particularly brown trout (*Salmo trutta*), was invariant with prey biomass across webs (Fig. S4). These results are consistent with previous work in these sites which has highlighted the importance of terrestrial prey for subsidizing the biomass production of these apex predators[30,31]. Deviations from the expected general scaling pattern could therefore be valuable for identifying the importance of environmental factors that permit some species an 'escape' from the predator-prey power law (see also[32]), and offers promising avenues for future research.

Our study, which takes a first step towards investigating predator-prey biomass scaling in complex food webs, has some notable limitations. First, information on the weighting of feeding links was not available for the food webs studied here, and a more comprehensive investigation should require specific interactions strengths and vulnerabilities of each species, data that is, as yet, unavailable. Although our results are robust to alternative assumptions in how these factors are treated (Table S5), any systematic variation in feeding interactions could play an important role. Second, information on the biomass of all basal resources was also not generally available, so our analysis focused on higher trophic predators feeding on (animal) prey. While

our approach may equally apply more generally to consumers and resources (e.g. aquatic snails feeding on biofilm), further work is required to test the generality of the empirical patterns we observed using more detailed datasets where this information, and data on interaction strengths, is widely available.

Overall, our study reveals a consistent sub-linear predator-prey scaling regime in complex food webs and makes a strong case for the existence of a systematic form of density-dependent population growth that governs the biomass structure of freshwater, marine and terrestrial ecosystems. The highly conserved predator-prey scaling we observed within and across food webs implies a relatively simple scaling-up of predator and prey population biomasses across levels of biological organization. These general patterns in energy flow between predator and prey could facilitate improvements in modelling trophic structure and community dynamics, as well as the corresponding ecosystem functions[4,5]. Our findings suggest sub-linear predator-prey biomass scaling holds within complex omnivorous food webs, urging ecologists to understand the origin of this large scale, cross-system pattern.

## Methods

We collated available food web datasets from a global database of traits and food-web architecture (GATEWAy v.1.0;[26]), where biomass and trophic interaction data were available across a large biomass gradient. Datasets for three ecosystem types met these requirements: UK freshwater streams[30,31], a global compilation from marine intertidal rock pools[25,33] and terrestrial soils of European forests[34,35] (Table S1).

We use the terms predators and prey throughout rather than the more general terms consumers and resources since information on the biomass of all resources is typically incomplete in food web studies (e.g. biomass of detrital matter in soils or macrophyte biomass in aquatic systems). While our approach may equally apply to primary consumers and their resources (e.g. aquatic snails feeding on biofilm), data limitations meant we restricted our analysis to biomass scaling patterns among ectothermic invertebrate and vertebrate predators and prey. This was achieved by filtering the data to include only predators that had a prey averaged trophic level[36] >2.5. To ensure robust power-law fits to the data, we excluded food webs that had fewer than five predators after this cut-off was applied, and where prey biomass varied by less than one order of magnitude. This resulted in 30 stream, 66 rock pool and 45 soil food webs for further analysis. Changing the cut-off value (e.g. to include predators with a trophic level > 3) yields similar sub-linear scaling exponents (Table S5). It does, however, result in generally greater variation in the 95% confidence intervals around the exponent estimates (Table S5), and lower ecosystem-level exponents estimates in the rock pool data, due, most likely, to the lower number of observations included in the analysis and reduced statistical power.

### Within-web scaling

To investigate biomass scaling within webs (Fig. 1c), we first calculated the biomass of each node in a food web as the numerical abundance (per m$^2$) multiplied by species average body mass (g dry weight). We then used the feeding link information to identify the prey items for each predator. Because more than one predator typically feeds upon a given prey species, each predator can therefore only consume a fraction of prey biomass production. Information on the weighting of feeding links (e.g. by the proportion of prey items found in predator's guts) was not available for the food webs studied here so interaction strengths were assumed to be equal. Specifically, we corrected the available biomass of prey for a given predator by considering that prey biomass has to be shared among all of its predators. Therefore, we divided the biomass of each prey node by its vulnerability (i.e. by the number of nodes which feed upon it). We then summed (vulnerability adjusted) prey biomass for all prey of a given predator. This approach assumes there is no overcompensation in the prey following predation

and thus no indirect facilitation among predators. Analyses not accounting for prey vulnerability yielded similar mean within- and across-web scaling exponents ($\bar{k}$ = 0.76 [CI: 0.68 to 0.83] and $\bar{k}$ = 0.68 [CI: 0.53 to 0.83], respectively), although within-web scaling in terrestrial webs was more sensitive to the prey vulnerability assumption than scaling relations in freshwater and marine webs (Table S5). Predators aggregated to coarse taxonomic groupings (e.g. zooplankton) were excluded from the within-web analysis (but included in the across-web analysis, described below). This resulted in the exclusion of 25 marine food webs (from the initial 66) for the within-web analysis, since the remaining number of consumers in the web was < 5.

We used linear mixed-effects (LME) modelling[37,38] to characterise within-web scaling using the lme4 package in R v. 3.0.2[39]. This modelling approach is appropriate given the multi-level structure of our data (e.g. the variance for each web is nested within that of the whole dataset) and its unbalanced nature (e.g. there is variation in the number of webs across ecosystem types)[37,38]. Furthermore, we expect the power-law exponent ($k$, slope), and the constant ($c$, intercept) to vary between-samples, $i$, (i.e. between food webs) due to factors such as species richness or resource availability, that affect community dynamics. LME modelling allows us to account for this variation by treating the slopes and intercepts as random variables with averages of $\bar{k}$, and $\bar{c}$, and deviations from these averages among samples of $\varepsilon_{k,i}$, and $\varepsilon_{c,i}$, respectively.

We used a top-down approach, starting with the most complex model, to determine the significance of the fixed and random effects terms in LME models in two-stages. We first determined whether it was necessary to include random effects corresponding to variation in both the slope and intercept among food webs. We did this by fitting a full model which included the fixed effects of log$_{10}$-transformed prey biomass, ecosystem type (freshwater, marine or terrestrial) and their interactions with the random effects of web identity on the slope and intercept (with correlation). We then compared this full model to simpler random-effects structures, which included the random effects of web identity on the slope and intercept and the random effect of web identity on intercept only. Comparison of AIC scores revealed that the random-effects structure that best described the data included random variation in the power-law exponent ($k$) and constant ($c$) attributable to web identity, with a correlation term (see Table S3). In the second stage, we applied the optimum random-effects structure determined in stage one and determined the significance of the fixed-effects by comparing models with and without an interaction term between ecosystem type and prey biomass (i.e. $k$) and a simpler model with prey biomass only (Table S3). A likelihood ratio test was used to assess the improvement in model fit[37,38]. The final model was then refitted using restricted maximum likelihood to determine the average parameter estimates of interest: $\bar{k}$, and 95% confidence intervals.

### Predator traits

In an additional analysis, we included predator omnivory and predator-prey body mass ratio as additional fixed effects in the final LME model for within-web scaling (described above). We considered additional variables such as predator generality, vulnerability and trophic level, but these were highly correlated ($r > 0.7$) with other predictor variables and so were not included in the statistical models. In our initial analysis, we were interested in comparing our results to Hatton et al.[8] where the sole effect of prey biomass was considered. This second set of models allowed us to assess the influence of additional factors on predator biomass and whether the importance of these varies between ecosystem types. Predator omnivory was calculated as the variance of the trophic level of the set of prey species, providing an omnivory value for each species within a web. The trophic level of species $i$ ($TL_i$) was defined as one plus the average trophic level of its prey, trophic level of basal species being set to one[36]. Predator-prey body mass ratio (PPmR) was calculated from the mean body mass of the predator and

of their prey and was $log_{10}$-transformed. The significance of factors and their interactions (Table S4) were determined using Satterthwaite's method for approximating degrees of freedom for the $t$- and $F$-tests from the lmerTest package in R.

## Across-web scaling

To investigate scaling relations across webs (Fig. 1d), we calculated the total biomass of predators and combined biomass of prey within each food web. Analysis of covariance was used to estimate $k$ because the across-web analysis includes only a single data point for each food web, and therefore mixed-effects models were no longer necessary at this organisational level. To investigate if biomass scaling was associated with changes in prey size structure, we re-ran the models replacing predator biomass with mean prey body mass and determined $k$ for the relationship of mean prey body mass (g) vs. total prey biomass (g/m$^2$). Four outliers were removed in this analysis (Table S1): these were streams where fish were naturally absent, leading to greatly different mean prey body mass estimates.

## Regression method

Currently, there is an ongoing debate about which regression methods are least biased depending on the distribution of error between x- and y-axis variables in fitting bivariate power laws in biology[40–42]. Here we used ordinary least square regression (OLS; type 1 regression) throughout to allow comparison to previous empirical syntheses[8,17,18]. We also adopted this approach because this likely represents the least biased slope estimator for the specific data that we report, given the greater fraction of error associated with estimating the biomass of predators which are rarer than their smaller prey.

## Reporting summary

Further information on research design is available in the Nature Research Reporting Summary linked to this article.

## Data availability

Data was obtained from a global database of traits and food-web architecture (GATEWAy v.1.0; https://idata.idiv.de/ddm/Data/ShowData/283?version=3). Processed data are available at https://figshare.com/s/49f09c604b5be6df7838.

## Code availability

The accompanying analysis R code is available at https://figshare.com/s/49f09c604b5be6df7838.

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

## Acknowledgements
The conception of this study emerged from the FuSED iDiv workshop, which was funded by the German Research Foundation (FOR 1451).

## Author contributions
U.B., B.G. and D.M.P. conceived the study design. All authors gathered, contributed or organized data. D.M.P, B.G, B.R and I.A.H. carried out statistical analyses. D.M.P. and I.A.H. made the figures. D.M.P. wrote the first draft of the manuscript. All authors discussed the results and commented on the manuscript.

## Funding

## Competing interests
The authors declare no competing interests.
