## [Peer Review File · Nature Communications]

Reviewer comments, first round -

Reviewer #1 (Remarks to the Author):

Review of the manuscript "Consistent predator-prey biomass scaling in complex food webs" for Nature Communications NCOMMS-21-38239 (pls see attached file)

This study aims to describe the pattern of how biomass of predators scales in relation to the biomass of their prey, and whether predator-prey body mass ratio and/or omnivory can explain the scaling. This is addressed by estimating the predator:prey biomass relationship in and across 141 different food webs of three different types (from soil, intertidal rocky shore, and stream ecosystems; denoted 'terrestrial', 'marine', and 'freshwater' in the paper), using data on food web structure (abundance, feeding links, and species mean weight) from a global database previously published by partly the same authors.

The question of biomass distribution in food webs dates back to Elton (1927) and is of general interest in ecology, as it can manifest energy flow (or efficiency) as well as responses to human pressures (e.g. fishing down the foodweb; Pauly et al. 1998). The particular pattern addressed here whether changes in predator biomass scales different from 1:1 with the biomass of their prey, is however not new. This was found previously by Hatton et al. (2015 Science), who is also a co-author of the submitted study. The authors point out that their analyses of predator-prey biomass scaling in a food web setting rather than for food chains is novel. This may be true, but does not come across very clearly when comparing the submitted study and Hatton et al (2015). Further details on the differences between the current study and earlier work, and the motivation for the current study is therefore needed.

That said, the authors find some interesting and important results, in particular at the level of 'populations'. The authors find a sublinear (i.e. less than 1:1-) scaling of predator to prey biomass for individual predator populations in all three ecosystem types, for predator species across multiple populations in different ecosystems (which they call 'metapopulations') and for total predator to total prey biomass within ecosystems. They find great variation in scaling exponent for individual predator populations between food webs, but no consistent difference between ecosystems, and thus an overall scaling exponent of 0.71. Importantly, the authors also found that predator-prey body mass ratio and degree of omnivory significantly increased the amount of variation in scaling of predator-prey biomass explained, but differently so depending on ecosystem type (PPMR in intertidal rocky shores, and omnivory in soils).

In contrast, the authors found a different scaling of predator to prey biomass among ecosystem types, when analyzed per predator species across multiple food webs (denoted 'metapopulations'), with values 0.07, 0.58 and 0.69, and with great variation among predator species. Regarding the scaling of total predator biomass to total prey biomass within a food web, there was in contrast no significant difference across the different types of ecosystem, for which the scaling exponent was 0.66. None of the patterns at these latter two levels of analyses were tested for the influence of any explanatory variables.

The authors conclude that there is a consistent scaling of predator to prey biomass across ecosystems and across these three levels of biological organization, and that it is close to 0.75. They highlight only one exception (predator species analyzed across foodwebs in soil ecosystems), which they put down to the importance of competition and habitat availability. They further go on to test whether prey size-structure can explain prey biomass, but find no relation and infer that prey biomass increases therefore stem from changes in prey density. They argue that this shows that density-dependent population growth [in prey] is the systematic cause of the predator-prey biomass scaling. Finally they claim that this finding can improve management of ecosystem services.

While the fundamental question of predator-to-prey biomass scaling is not novel, the findings at the population level are important, especially of factors that contribute to explaining variation in

that scaling among predator populations is a significant contribution. Some of the conclusions made, however, are not fully supported by the results (e.g. the statement that there is a near-0.75 scaling that holds across biological organization and types of ecosystem).

I also have some concerns regarding the logic of the presentation. First, while the analysis at the level of populations is straight forward, as is that of the total predator to total prey biomass ('the ecosystem level'), I fail to see what the intermediate level represents and why it is relevant (in addition, terming this metapopulations is inappropriate, as the species in different ecosystems are not shown herein to be linked through dispersion). Second, the additional analyses introduced in the discussion comes across as post-hoc, and are not introduced or motivated in the introduction. Rather than glossing over this in the introduction (lines 73-74), I think the reader would be much helped if you explained the proposed mechanism in the introduction, and let that theory guide and motivate the set of analyses and the subsequent discussion.

I also have some (smaller) questions regarding the methods, specified in detailed comments below.

Because the authors set out to study whether the sublinear scaling of predator to prey biomass found in earlier studies can be generalized to food webs, it surprises me that there is no discussion of how general their findings may be (e.g. is there really a general scaling when some analyses show 0.58 and others 0.71? or when there is such great variation around it?), especially in relation to the datasets chosen. I lack a critical discussion of this, in particular because the studied systems are limited to soils, intertidal rocky shores and streams (whereas the authors tend to overemphasize the generality by renaming these as terrestrial, marine and freshwater). For example, is this likely to hold for open-water (pelagic) food webs? Or above ground terrestrial food webs? Why/why not? The paper also contains several formulations overstating the implications of the findings, without providing any specification or clarifying example; statements such as that the findings "could facilitate improvement in management of ecosystem services" needs to be exemplified or explained to be convincing.

Overall, I think the main question addressed is interesting to a wide audience in ecology, and some of the findings could provide an important contribution to the field, but the interpretation of the results and the presentation requires substantial work to provide a logical structure and thorough discussion.

Detailed comments:

Logical structure: to me it seems you could ask 1. what is the biomass scaling in foodwebs, 2. what causes the mean scaling and 3. what causes variation around it. And then answer it using your population-level and ecosystem-level analyses, the analyses that you've currently put in the discussion, and then the analysis of the effect of PPRM and omnivory (all in this order – currently they are spread out in the paper, and interrupted by the 'metapopulation'-level analyses). Rather than asking whether a previously found pattern can be found again, in more complex systems. That would give you a more theory-guided set-up of the study.

line 65: please explain what you mean by "greater ecosystem-level biomass structure" (how can a structure be greater?)

lines 73-74: please explain why a sublinear scaling "suggests a systematic form of density-dependent population growth". how does this work? what are the mechanisms? and how could these be tested? (the analyses you introduce in the discussion suggests that a sublinear scaling is not enough to indicate "a systematic form of density-dependent growth that structures biomass distribution in food webs)

lines 93-94: this type of statement calls for that you would later in the discussion) exemplify how this would be done, based on your results (or, explain how it would work in the introduction, immediately following these lines)

lines 106: if you are planning to keep the 'meta-population' level, I strongly suggest that you change the term for it; a metapopulation is a very specific thing, where local populations are connected via dispersal. What you are analyzing are species in different food webs, not populations that are connected via dispersal.

line 137: you have very wide Cis around your estimates (e.g. Fig 2 bottom row shows they are

non-overlapping in some cases); I agree that you can conclude that there may be a common scaling, as you do not find a significant effect of including ecosystem-specific scaling exponent. But I think you need to acknowledge the visible great variation around it. For example, by referring also to Fig. 2 on line 147-149 where you nicely and correctly point out the notable variation.

line 161-174: this is a very good, useful and important analysis!

line 187: please rephrase (as a minimum, remove 'i.e.' as what follows isn't the definition of a metapopulation – see comment on this terminology, above)

line 214: suggest you remind the reader about the motivation for this level of the analyses, with a sentence

line 220: remove the double 'test'

line 240: a statement such as "...than previously recognized" requires citations to those studies that 'previously have recognized'

line 252: why would competition and habitat availability be more important in soil ecosystems than in intertidal rocky shores or streams?

lines 276-277: why would predator biomass and prey productivity be linearly related? doesn't predator functional responses of type II and type III (as commonly found), imply that they would be sub-linearly related?

line 278-281: please clarify how you mean this would work; your point (1) applies to individuals, but you use it to talk about processes limiting populations?

lines 292-293: this is really important finding of great significance to the field – I suggest you give the underlying analyses more emphasis and discuss these thoroughly (e.g. continue on line 297 to give alternative explanations to why the factors have different in the intertidal rocky shore compared to the soil ecosystems; is it likely that this difference would hold when looking also at other types of terrestrial ecosystems and other types (and truly) of marine ecosystems?)

line 309: please explain how this would be done

line 334-335: is this really true? Table S5 seems to suggest that >3TL does have a very large influence on e.g. the 'marine' ecosystems

line 347-350: would be good to point out that by this approach you assume that there are no overcompensation in the prey following predation, and thus no indirect facilitation among predators

line 351-352 & Supplement, Table 5: how can results from the main text and those from ignoring prey vulnerability be considered similar when Table S5 report the exponents to be 0.75 and 0.98 respectively (for 'terrestrial' ie. soil ecosystems), with CIs that don't overlap? (Similarly, at the ecosystem level the exponents in the marine ecosystems seem very sensitive to the prey vulnerability assumption, as their CIs barely overlap)

Supplement, Table 2: suggest you rephrase the final sentence of the legend as follows, for clarity: "...reveal that the simpler F1 was not a significantly better fit to the data than model F2 (at..) and that the more complex F3 was a significantly worse fit than model F2."

Supplement, Fig. S7: please correct the beginning of the figure legend; you cannot talk about prey size structure when what you are testing is prey mean size (size structure can vary greatly, despite having identical mean size). This is also very important to correct and be specific on when discussing it in the main text (currently only in the discussion).

Reviewer #2 (Remarks to the Author):

I have read the paper by Perkins and collaborators where they quantify the relationship between predator and prey biomass at three different levels of data aggregation (within food webs, across food webs, and across ecosystems). Their results suggest that, as also found for pairwise interactions, regardless of whether the data is aggregated within food webs, across, or by ecosystem type, the biomass scaling between predators and prey is roughly 3/4. As per usual from these authors, the paper is very well written, in general, and I find the findings to be of interest as well. I make suggestions below that I hope will help the authors clarify some aspects of their manuscript.

MAJOR

1) My main problem is that it isn't clear at all why we would expect anything other than a 3/4

scaling law in this analysis. All pairwise interactions are indeed part of a food web somewhere, so at the very least we should actually expect a $3/4$ scaling law within food webs (what the authors call "population level", see my comment below). What would be interesting, perhaps, is for the authors to explain whether we expect the scaling to differ from $3/4$ at a different level of aggregation, and why. In other words, the paper is framed as a hypothesis-free exploration, and given the paper by Hatton et al, and given the overlap in authors, I was expecting an analysis that was more grounded in theory or existing data to at least come up with hypotheses/expectations for each level of aggregation. My suggestion is that you, therefore, provide those hypotheses, either from first principles, or from theory, as to why we would expect/would not expect those scaling to be different across the levels of analysis. That way you help your reader understand what is interesting about these results, especially in light of Hatton et al 2015.

2) I don't think that the definitions of the different levels of analysis ("population level", "metapop" and so on) make much sense. I see what the authors are trying to do here, I just think the names are incorrect. At the population level, the data point is actually a property of the predator-prey pair, not of any one population on its own. At the metapopulation level, there isn't an actual metapopulation component, at least as understood from classic metapopulation theory (collection of local populations connected by dispersal), nor would "metacommunity" be appropriate either since there is, as far as the authors can tell (and unless I'm missing something), actual dispersal between these food webs, or their data is blind to these effects. I think the only name I can get behind is the ecosystem one. All options I can think of are a bit verbose, but probably better than current choices: "intra food web", "inter food web" and "inter ecosystem" comparisons or something like that.

NOTE: I retrospect, it was clear from continuing my read that the authors do have information on taxa that occur in multiple food webs. Whether these actually represent true meta-populations is a different issue, and would need to be better explained and discussed. Still, I argue that no relationship shown here occurs at the population level as for any slope to exist data is needed from more than one population, and regardless, any given predator-prey interaction is already occurring between at least two populations. Anyhow, this is just a wordsmithing issue.

3) Not clear what is meant by "we constructed predator-prey relationships". How do you construct those or what does that actually mean? This being central to understanding the methods, I believe this should be well explained in the main text.

4) I feel like a bit more handholding would be needed for the readers to understand where some of this variation may be coming from. For example, both freshwater and marine ecosystems show some negative slopes (smaller predator relative to prey!) but this doesn't show up in terrestrial systems. Yet, aquatic systems being gape limited, I would have hypothesized that negative slopes should not be found in such systems, but could occur in terrestrial systems (e.g., spiders eating birds, lions consuming elephants, while rare, it does occur). I suggest the authors 1) make some of the tested hypotheses more explicit so the results don't come out of the blue when presented, and at least discuss why some of the observed patterns make/doesn't make sense based on what we know about the way predator prey interactions occur in different ecosystems.

MINOR

ABSTRACT

L-38: No need to state it's powerful. It can be powerful and it also has limitations.

L-46: I think stating it's biomass matters. I suggest "more prey biomass supports proportionally less predator biomass", otherwise you need to say "fewer predators" instead of less, but then the analyses weren't done on abundances, which is why I suggest the phrasing above.

INTRO

L-64: Depends on how you define recent. 1yrs? 2yrs? 5yrs? Then no. Last decade? Then yes. But is within 10 years that recent? Definitely not dumping on Hatton et al, great paper and definite citation here. I'd just say a "previous" general finding since "recent" is on the eyes of the beholder.

DISCUSSION:

L-292: what do you mean flow is enhance to highly omnivory taxa? You mean "by" highly

omnivory taxa? That omnivores enhance flow within food webs is trivially true and this analysis wasn't needed to say that given that omnivores prey upon multiple trophic levels, thus curt-circuiting intermediate TLs.

L-308-311: I think that going from your results to these implications is a major overreach.

note: all Science papers in your reference list have (80-) right after the name if the journal. I get it, nobody cares, but I thought you should know your reference editor is doing this consistently.

Reviewer #3 (Remarks to the Author):

This is an high-quality manuscript that will be of considerable interest to a broad readership. The findings are novel, it is very well written and the analyses are robust. I have only 2 general points of (constructive) criticism:

- 1) First, it is a pity that connections have not been drawn with theoretical and empirical work on size spectra given that much of this work has a similar focus to this manuscript and offers highly complementary insights;
- 2) Second, the authors seem to implicitly assume that rock pools are broadly representative of marine ecosystems in general. This is not the case. Rock pools are essentially 2-dimensional and have strong system boundaries, and are more akin to streams and terrestrial foodwebs in these important respects. It is well-recognised that both dimensionality and 'open-ness' have important implications for food web structure and function and this needs to be recognised and considered in the discussion. It would be very interesting to also briefly hypothesis whether and how these relationships might change in other marine ecosystems.

I believe that addressing these points will considerably improve what is already a good paper by broadening its relevance and better contextualizing it within a richer body of theoretical and empirical work.

In addition to these general points, I offer some specific suggestions:

- line 56: consider adding "and individuals" after "species" in recognition of the fact that in many marine taxa, trophic position is more strongly tied to individual body size (and life history stage) than to species. The importance of ontogenetic omnivory in indeterminate growers could be better considered throughout.
- line 78: as above, I suggest also mentioning ontogenetic omnivory here as this is a key feature of marine food webs.
- lines 88-94: work on biomass spectra has been exploring these concepts for the better part of 50 years, and would warrant mention here.
- Figure 2-5 and methods: it wasn't entirely clear to me why the 3 ecosystem types were consistently modeled separately. Was there a strong reason for doing this instead of including ecosystem type as a fixed effect (with an interaction with prey biomass) in the models? This would give you an estimate of the overall cross-ecosystem slope (and arguably a more parsimonious way of doing this than by considering the average of the individual slopes) and also better enable comparing slopes and intercepts across ecosystems.
- line 239-240: I'm not sure that the statement regarding greater consistency in biomass structure than had been previously recognized is true for marine systems where consistency of biomass spectra has been appreciated for decades.
- Lines 261-263: could draw parallels with work examining 'escapes' from size spectra arising from subsidies here
- Lines 311-313: I suggest tempering this final statement somewhat. Perhaps reword to "provides insight into baselines for structure...". I'm skeptical that the insights from rockpools shed light on baselines for marine ecosystems more broadly.

Papers that may be helpful to reference in addressing the above:

- Andersen KH, Berge T, Gonçalves RJ, Hartvig M, Heuschele J, Hylander S, Jacobsen NS, Lindemann C, Martens EA, Neuheimer AB, Olsson K, Palacz A, Prowe AEF, Sainmont J, Traving SJ, Visser AW, Wadhwa N & Kjørboe T (2016). Characteristic Sizes of Life in the Oceans, from Bacteria to Whales. *Annual Review of Marine Science* 8(1):217–241.

- Pawar S, Dell AI, & Van M. Savage (2012). Dimensionality of consumer search space drives trophic interaction strengths. *Nature* 486(7404):485–489.
- Jennings S, Mélin F, Blanchard JL, Forster RM, Dulvy NK & Wilson RW (2008). Global-scale predictions of community and ecosystem properties from simple ecological theory. *Proceedings of the Royal Society B: Biological Sciences* 275(1641):1375–1383.
- Polishchuk LV & Blanchard JL (2019). Uniting Discoveries of Abundance-Size Distributions from Soils and Seas. *Trends in Ecology & Evolution* 34(1):2–5.

We would like to thank all three reviewers for their insightful comments, which have greatly helped improve the manuscript. We are confident that we have fully addressed all their comments. In summary, we have made the following changes:

- Revised the presentation and discussion of the results, providing a better manuscript structure
- Revised the terminology in line with suggestions by the reviewers
- Added a dedicated section linking biomass scaling patterns to current theoretical frameworks (Text box 1)
- Provided new text on linking biomass distributions across levels of organization (Appendix S1 in Supplementary Material)
- Made minor changes to the manuscript format so that it complies with the required format for Nature Communications as detailed in the checklist.

We provide a point-by-point response to reviewers' comments below.

REVIEWER COMMENTS

Reviewer #1 (Remarks to the Author):

Review of the manuscript "Consistent predator-prey biomass scaling in complex food webs" for Nature Communications NCOMMS-21-38239 (pls see attached file)

This study aims to describe the pattern of how biomass of predators scales in relation to the biomass of their prey, and whether predator-prey body mass ratio and/or omnivory can explain the scaling. This is addressed by estimating the predator:prey

biomass relationship in and across 141 different food webs of three different types (from soil, intertidal rocky shore, and stream ecosystems; denoted 'terrestrial', 'marine', and 'freshwater' in the paper), using data on food web structure (abundance, feeding links, and species mean weight) from a global database previously published by partly the same authors.

The question of biomass distribution in food webs dates back to Elton (1927) and is of general interest in ecology, as it can manifest energy flow (or efficiency) as well as responses to human pressures (e.g. fishing down the foodweb; Pauly et al. 1998). The particular pattern addressed here whether changes in predator biomass scales different from 1:1 with the biomass of their prey, is however not new. This was found previously by Hatton et al. (2015 Science), who is also a co-author of the submitted study. The authors point out that their analyses of predator-prey biomass scaling in a food web setting rather than for food chains is novel. This may be true, but does not come across very clearly when comparing the submitted study and Hatton et al (2015). Further details on the differences between the current study and earlier work, and the motivation for the current study is therefore needed.

That said, the authors find some interesting and important results, in particular at the level of 'populations'. The authors find a sublinear (i.e. less than 1:1-) scaling of predator to prey biomass for individual predator populations in all three ecosystem types, for predator species across multiple populations in different ecosystems (which they call 'metapopulations') and for total predator to total prey biomass within ecosystems. They find great variation in scaling exponent for individual predator populations between food webs, but no consistent difference between ecosystems, and thus an overall scaling exponent of 0.71. Importantly, the authors also found that predator-prey body mass ratio and degree of omnivory significantly increased the amount of variation in scaling of predator-prey biomass explained, but differently so depending on ecosystem type (PPMR in intertidal rocky shores, and omnivory in soils).

In contrast, the authors found a different scaling of predator to prey biomass among ecosystem types, when analyzed per predator species across multiple food webs (denoted 'metapopulations'), with values 0.07, 0.58 and 0.69, and with great variation among predator species. Regarding the scaling of total predator biomass to total prey biomass within a food web, there was in contrast no significant difference across the different types of ecosystem, for which the scaling exponent was 0.66. None of the patterns at these latter two levels of analyses were tested for the influence of any explanatory variables.

The authors conclude that there is a consistent scaling of predator to prey biomass across ecosystems and across these three levels of biological organization, and that it is close to 0.75. They highlight only one exception (predator species analyzed across foodwebs in soil ecosystems), which they put down to the importance of competition and habitat availability. They further go on to test whether prey size-structure can explain prey biomass, but find no relation and infer that prey biomass increases therefore stem from changes in prey density. They argue that this shows that density-dependent population growth [in prey] is the systematic cause of the predator-prey biomass scaling. Finally they claim that this finding can improve management of ecosystem services.

While the fundamental question of predator-to-prey biomass scaling is not novel, the findings at the population level are important, especially of factors that contribute to explaining variation in that scaling among predator populations is a significant contribution. Some of the conclusions made, however, are not fully supported by the results (e.g. the statement that there is a near-0.75 scaling that holds across biological organization and types of ecosystem).

I also have some concerns regarding the logic of the presentation. First, while the analysis at the level of populations is straight forward, as is that of the total predator to total prey biomass ('the ecosystem level'), I fail to see what the intermediate level represents and why it is relevant (in addition, terming this metapopulations is inappropriate, as the species in different ecosystems are not shown herein to be linked through dispersion).

We would like to thank the reviewer for the thorough assessment of the manuscript. Regarding this first point, we agree that it is not straightforward to link the intermediate scaling relation to both the population and ecosystem relations. We have therefore decided to remove this analysis from the main text. We highlight in the discussion how this approach could still be informative (page 15, lines 857-864), and provide some descriptive analysis in supplementary (Figure S4), but we have removed the term 'metapopulation-level scaling' throughout.

Second, the additional analyses introduced in the discussion comes across as post-hoc, and are not introduced or motivated in the introduction. Rather than glossing over this in the introduction (lines 73-74), I think the reader would be much helped if you explained the proposed mechanism in the introduction, and let that theory guide and motivate the set of analyses and the subsequent discussion.

Thank you for this suggestion. We now provide a more detailed motivation for looking at this relationship in the introduction (Text box 1) and comment fully on the results of this analysis in the Results.

Text box extract:

“Theoretical predictions for biomass scaling

Many strands of ecological theory provide predictions for the distribution of biomass between predator-prey pairs and along food chains. However, we are not aware of any theory that makes predictions consistent with the sub-linear scaling of the predator-prey power law (Fig. 1a).

Predator-prey theory models *the dynamics of feeding interactions, and has traditionally focused on two distinct trophic levels, rather than on networks of highly omnivorous food webs¹¹. For a given model, we relate the equilibrium predator and prey biomass across a simulated gradient of enrichment, and ask whether it recovers the empirical scaling observed. A biomass or 'enrichment' gradient can be modeled by varying a parameter in the prey growth term (Table S1), which may be exponential or logistic, or else in the predator-prey interaction term, which may be a linear or saturating functional response^{8,11,12}. In Fig. Box 1, we show equilibrium predictions for three classic models that are not consistent with predator-prey scaling, either because they are not stable^{13,14} (a), do not provide meaningful relations^{11,12} (b) or predict an exponent of 1^{15,16} (c).*

We can obtain consistent sub-linear biomass scaling by setting prey growth to have the same exponent as predator-prey biomass scaling^{8,13-16} (Fig. Box 1 d). Sub-exponential scaling of prey production with biomass (with an exponent of $\sim 3/4$), stabilizes simple models such as Lotka-Volterra, and gives similar ($\sim 3/4$ power) predator-prey equilibrium scaling. Two possible reasons explain sub-linear prey productivity scaling (1) allometric constraints on individual rates which scale allometrically with body mass and could lead to sub-linear community scaling with a systematic relation of body size to biomass¹⁷, and/or (2) density dependent effects - i.e. submaximal individual growth⁸. Hatton et al.⁸ found that mean prey body mass varied little over the prey biomass gradient and thus their results are consistent with the role of density dependent effects on population growth. Alternatively, we could introduce 'interference scaling' in the functional response or density dependence in the predator mortality (exponent $\sim 1/3$). Despite the various possibilities, there is, as yet, no firm theoretical basis on which to understand sub-linear biomass scaling.

Fig. Box 1 | Simple predator-prey model predictions across simulated biomass gradients. Gradients are simulated by varying interaction strength in a) and d), and by varying a logistic carrying capacity in b) and c). Models a) to c) do not make predictions consistent with empirical sublinear predator-prey biomass scaling. Model d) makes realistic predictions but requires that prey growth is sub-exponential, which is an assumption that requires new theory. Predator-prey model equations used to generate the phase space are provided in Table S1.”

I also have some (smaller) questions regarding the methods, specified in detailed comments below.

Because the authors set out to study whether the sublinear scaling of predator to prey biomass found in earlier studies can be generalized to food webs, it surprises me that there is no discussion of how general their findings may be (e.g. is there really a general scaling when some analyses show 0.58 and others 0.71? or when there is such great variation around it?), especially in relation to the datasets chosen. I lack a critical discussion of this, in particular because the studied systems are limited to soils, intertidal rocky shores and streams (whereas the authors tend to overemphasize the generality by renaming these as terrestrial, marine and freshwater). For example, is this likely to hold for open-water (pelagic) food webs? Or above ground terrestrial food webs? Why/why not? The paper also contains several formulations overstating the implications of the findings, without providing any specification or clarifying

example; statements such as that the findings “could facilitate improvement in management of ecosystem services” needs to be exemplified or explained to be convincing.

Thank you for this comment. We agree that there is some uncertainty as to whether our results would extend to other ecosystem types. We have now provided a paragraph in the discussion commenting on whether a similar pattern might hold in pelagic food webs, where ontogenetic diet shifts are common.

New text, page 14, line 350: *“Further research would be instructive to understand if these are general patterns across different types of terrestrial and aquatic ecosystems. For instance, whilst rock pool food webs display very similar topological organization and PPMR scaling as open marine webs^{35,36}, we might expect different scaling patterns in pelagic marine webs where trophic interactions take place in three-dimensions³²; where ontogenetic diet shifts are common³⁷; and where food chains are long²⁶. Adapting our food-web approach to quantify biomass scaling among size classes would likely be informative for tackling this”.*

Overall, I think the main question addressed is interesting to a wide audience in ecology, and some of the findings could provide an important contribution to the field, but the interpretation of the results and the presentation requires substantial work to provide a logical structure and thorough discussion.

Thank you for your positive opinion of the manuscript. We too believe this work will appeal to a wide audience and believe, with the help of the reviewers’ suggestions, we have improved the interpretation, presentation and discussion of the results and manuscript structure.

Detailed comments:

Logical structure: to me it seems you could ask 1. what is the biomass scaling in foodwebs, 2. what causes the mean scaling and 3. what causes variation around it. And then answer it using your population-level and ecosystem-level analyses, the analyses that you’ve currently put in the discussion, and then the analysis of the effect of PPRM and omnivory (all in this order – currently they are spread out in the paper, and interrupted by the ‘metapopulation’-level analyses). Rather than asking whether a previously found pattern can be found again, in more complex systems. That would give you a more theory-guided set-up of the study.

We would like to thank the reviewer for this very helpful comment. We have followed the suggested structure and believe this has greatly improved the manuscript. For instance, we now outline the research questions (in the order suggested by the reviewer) in the introduction and revisit these in the same order in the results. New text, page 7, line 185: *“Specifically, we ask: (i) how does predator and prey biomass scale in complex food webs? (ii) what role does omnivory and variation in the size of predators relative to their prey have on the scaling pattern? (iii) does a general power-law hold across levels of biological organization? and (iv) is the association between predator and prey biomass underpinned by changes in prey density or the average size of prey? Our results reveal fundamental similarities in the biomass structure of food webs in very different ecosystem types”*

line 65: please explain what you mean by “greater ecosystem-level biomass structure” (how can a structure be greater?)

We agree this statement was a little vague. We have replaced “*greater ecosystem-level biomass structure*” with “*remarkable regularity in how the distribution of biomass across populations responds to enrichment*” (page 2, line 65).

lines 73-74: please explain why a sublinear scaling “suggests a systematic form of density-dependent population growth”. how does this work? what are the mechanisms? and how could these be tested? (the analyses you introduce in the discussion suggests that a sublinear scaling is not enough to indicate “a systematic form of density-dependent growth that structures biomass distribution in food webs). In our initial submission we avoided a detailed discussion of how the previous results from Hatton et al. (2015) were consistent with a systematic form of density dependent growth to avoid disrupting the flow of the introduction. We now address how this works, the mechanisms at play, and how these can be tested for, in a new dedicated section (Text box 1). Please see an extract from this given in the response above.

lines 93-94: this type of statement calls for that you would later in the discussion) exemplify how this would be done, based on your results (or, explain how it would work in the introduction, immediately following these lines).

To aid the flow of ideas and avoid repetition, we have decided to remove this sentence from the introduction and instead expand upon this topic in the discussion.

New text, page 15, line 361:

*“The regularity in predator-prey scaling we observed could provide insight into baselines for the biomass structure of natural communities, which could be informative for assessing the effects of environmental impacts within ecological communities and ecological status. For instance at the population-level, deviations away from these baselines in the form of smaller power-law exponents (shallower slopes) could reflect local perturbations (e.g. acidification, warming, over-exploitation) which have a disproportionate impact among larger organisms at higher trophic levels³⁸. Predator-prey biomass scaling could therefore offer a complementary approach to body size distributions and size spectra for evaluating ecosystem health³⁹. A similar approach could be applied for scaling relations within species, where the same species occur in multiple webs. Doing so could reveal how the biomass of a given predator species responds to variation in prey availability. For instance, among the stream food webs studied here, two common fish species displayed the characteristic near $3/4$ -power scaling pattern, whilst the biomass of salmonids, and particularly brown trout (*Salmo trutta*), was invariant with prey biomass across webs (Fig. S4). These results are consistent with previous work with these data which has highlighted the importance of terrestrial prey for subsidizing the biomass production of these apex predators^{40,41}. Deviations from the expected general scaling pattern could therefore be valuable for identifying the importance of environmental factors (such as cross-ecosystem subsidies) that permit some species an ‘escape’ from the predator-prey power law (see also⁴²), or that override the general scaling pattern at the population-level, and offers a promising avenue for future research”*

lines 106: if you are planning to keep the ‘meta-population’ level, I strongly suggest that you change the term for it; a metapopulation is a very specific thing, where local

populations are connected via dispersal. What you are analyzing are species in different food webs, not populations that are connected via dispersal.

Thank you for this suggestion. As mentioned in a previous response (above), we have decided to remove a formal analysis of this scaling relation from the main text and we now refrain from using the term ‘metapopulation’ (please see also the revised text above).

line 137: you have very wide CIs around your estimates (e.g. Fig 2 bottom row shows they are non-overlapping in some cases); I agree that you can conclude that there may be a common scaling, as you do not find a significant effect of including ecosystem-specific scaling exponent. But I think you need to acknowledge the visible great variation around it. For example, by referring also to Fig. 2 on line 147-149 where you nicely and correctly point out the notable variation.

We agree that there is notable variation around the mean scaling relationships and have amended the text in the results section to highlight this.

Amended text, page 8, line 212: *“Doing so revealed that the average exponent for each ecosystem type was similar, although confidence intervals were wide: freshwater $k = 0.61$ (95% CI 0.50 to 0.71), marine $k = 0.74$ (CI 0.66 to 0.82), and terrestrial $k = 0.75$ (CI 0.66 to 0.83) (Fig. 2; Table S3)”*.

Revised text, page 9, line 224: *“However, as indicated by the significance of the random-effects terms in the model (Table S4), there was notable variation in exponents between food webs – i.e. at local scales (Fig. 2).”*

line 161-174: this is a very good, useful and important analysis!

Thank you for this very positive comment! We also believe this analysis provides novel insights into energy flow in natural food webs.

line 187: please rephrase (as a minimum, remove ‘i.e.’ as what follows isn’t the definition of a metapopulation – see comment on this terminology, above).

In line with the comments addressed above, this text has now been removed from the manuscript.

line 214: suggest you remind the reader about the motivation for this level of the analyses, with a sentence.

Thank you for this suggestion. We have added the following sentence to remind the reader of this. Page 11, line 263: *“In order to investigate if similar predator-prey biomass scaling patterns emerge at the ecosystem-level...”*

line 220: remove the double ‘test’

Removed, thank you.

line 240: a statement such as “...than previously recognized” requires citations to those studies that ‘previously have recognized’.

Removed, thank you.

line 252: why would competition and habitat availability be more important in soil ecosystems than in intertidal rocky shores or streams?

In line with the comments addressed above, this text has now been removed from the manuscript.

lines 276-277: why would predator biomass and prey productivity be linearly related? doesn't predator functional responses of type II and type III (as commonly found), imply that they would be sub-linearly related?

Thank you for raising this important point. We have now expanded on the underlying theory behind sub-linear biomass scaling patterns in Text box 1 and have addressed this point specifically on page 3, line 89:

"A biomass or 'enrichment' gradient can be modeled by varying a parameter in the prey growth term, which may be exponential or logistic, or else in the predator-prey interaction term, which may be a linear or saturating functional response"

line 278-281: please clarify how you mean this would work; your point (1) applies to individuals, but you use it to talk about processes limiting populations?

We have modified the discussion text to clarify this point. Page 14, line 331:

"Despite the lack of any obvious mechanisms, we can make a steady state assumption that predator biomass is maintained in proportion to prey production^{8,34}. This would suggest that as prey biomass increases, their total productivity should scale near $\sim 3/4$ to match the predator biomass they support. Density dependent effects could cause per capita growth to decline sub-exponentially (Fig. Box 1 d) and force predators to generalize their consumption over a broader prey base (Fig. Box 2 g). Indeed, we observed that changes in prey biomass were primarily driven by changes in prey density, rather than average body size, consistent with density dependent effects driving the sub-linear nature of predator-prey biomass relations."

lines 292-293: this is really important finding of great significance to the field – I suggest you give the underlying analyses more emphasis and discuss these thoroughly (e.g. continue on line 297 to give alternative explanations to why the factors have different in the intertidal rocky shore compared to the soil ecosystems; is it likely that this difference would hold when looking also at other types of terrestrial ecosystems and other types (and truly) of marine ecosystems?).

Thank you for this positive comment and suggestion. As mentioned in the response above, we agree that there is some uncertainty as to whether our results would extend to other ecosystem types. We have now provided a paragraph in the discussion commenting on whether a similar pattern might hold in pelagic food webs, where ontogenetic diet shifts are common (see text pasted in response above; page 14, lines 350-356).

line 309: please explain how this would be done.

In revising the discussion text we have removed this sentence from the final paragraph. However, we now provide a detailed paragraph (page 15, lines 361-380, see also new text pasted in response above) outlining how our empirical relations could provide a baseline for understating ecological status of natural populations and communities.

line 334-335: is this really true? Table S5 seems to suggest that $>3TL$ does have a very large influence on e.g. the 'marine' ecosystems.

Thank you for raising this point. The scaling patterns are remarkably similar in most cases using a TL cut-off of 2.5 or 3. Thus the choice of approach does not alter the

main conclusions of this study. We do, however, now acknowledge explicitly the effect of TL cut-off on the ecosystem-level scaling patterns for rock pools in the revised text. Page 17, line 410: *“Changing the cut-off value, for example, to include predators with a trophic level > 3 yields similar sub-linear scaling exponents (Table S6). It does, however, result in generally greater variation in the 95% confidence intervals around the exponent estimates (Table S6), and lower ecosystem-level exponents estimates in the rock pool data, due, most likely, to the lower number of observations included in the analysis and reduced statistical power”*.

line 347-350: would be good to point out that by this approach you assume that there are no overcompensation in the prey following predation, and thus no indirect facilitation among predators.

Thank you for this suggestion. We have added this to the methods section.

line 351-352 & Supplement, Table 5: how can results from the main text and those from ignoring prey vulnerability be considered similar when Table S5 report the exponents to be 0.75 and 0.98 respectively (for ‘terrestrial’ ie. soil ecosystems), with CIs that don’t overlap? (Similarly, at the ecosystem level the exponents in the marine ecosystems seem very sensitive to the prey vulnerability assumption, as their CIs barely overlap).

Thank you for raising this point. We have added text to clarify this point by highlighting that, whilst the overlap in scaling relations can be considered similar to those with the vulnerability correction, there is notable variation within ecosystem types. Revised text, page 18, line 430:

“Analyses not accounting for prey vulnerability yielded similar mean population- and ecosystem-level scaling exponents to those we present in the main text ($\underline{k} = 0.76$ [CI: 0.68 to 0.83] and $\underline{k} = 0.68$ [CI: 0.53 to 0.83], respectively), although population-level scaling in terrestrial webs was more sensitive to the prey vulnerability assumption than scaling relations in freshwater and marine webs (Table S6)”.

Supplement, Table 2: suggest you rephrase the final sentence of the legend as follows, for clarity: “...reveal that the simpler F1 was not a significantly better fit to the data than model F2 (at..) and that the more complex F3 was a significantly worse fit than model F2.”.

Thank you for this comment. We have changed this table heading (now Table S4) in line with this suggestion.

Supplement, Fig. S7: please correct the beginning of the figure legend; you cannot talk about prey size structure when what you are testing is prey mean size (size structure can vary greatly, despite having identical mean size). This is also very important to correct and be specific on when discussing it in the main text (currently only in the discussion). We have replaced the term “prey size structure” with “mean prey size” in both the figure (now Figure 5) and the results and discussion text.

Reviewer #2 (Remarks to the Author):

I have read the paper by Perkins and collaborators where they quantify the relationship between predator and prey biomass at three different levels of data aggregation (within food webs, across food webs, and across ecosystems). Their

results suggest that, as also found for pairwise interactions, regardless of whether the data is aggregated within food webs, across, or by ecosystem type, the biomass scaling between predators and prey is roughly $3/4$. As per usual from these authors, the paper is very well written, in general, and I find the findings to be of interest as well. I make suggestions below that I hope will help the authors clarify some aspects of their manuscript.

Thank you for your positive opinion of the manuscript. We too believe this work will appeal to a wide audience and believe, with the help of the reviewers' suggestions, we have improved the interpretation, presentation and discussion of the results and manuscript structure.

MAJOR

1) My main problem is that it isn't clear at all why we would expect anything other than a $3/4$ scaling law in this analysis. All pairwise interactions are indeed part of a food web somewhere, so at the very least we should actually expect a $3/4$ scaling law within food webs (what the authors call "population level", see my comment below). What would be interesting, perhaps, is for the authors to explain whether we expect the scaling to differ from $3/4$ at a different level of aggregation, and why. In other words, the paper is framed as a hypothesis-free exploration, and given the paper by Hatton et al, and given the overlap in authors, I was expecting an analysis that was more grounded in theory or existing data to at least come up with hypotheses/expectations for each level of aggregation. My suggestion is that you, therefore, provide those hypotheses, either from first principles, or from theory, as to why we would expect/would not expect those scaling to be different across the levels of analysis. That way you help your reader understand what is interesting about these results, especially in light of Hatton et al 2015.

Thank you for this comment. We have now revised the introduction to explain our expectations (Page 6 line 163, page 6 line 175). In particular we highlight more clearly that variation in size structure, prey partitioning among predators and degree of omnivory could override the previously observed scaling pattern by Hatton et al. 2015. We now provide a dedicated section on linking scaling patterns across levels of organisation and provide this as an appendix in the supplementary. Here we have undertaken preliminary simulations where predator species share two or three prey species with other predators. Whilst this work suggests that population predator-prey scaling implies similar ecosystem predator-prey scaling in the limit of very simple systems, predictions are more involved in more complex food webs with variation in size structure, prey partitioning among predators and degree of omnivory. We therefore acknowledge that further theoretical work is needed to know if constancy or lack of systematic variation in these properties is sufficient to predict ecosystem level scaling from the aggregate of population level scaling. New text:

Appendix S1 | Linking ecosystem- to population-level biomass scaling

Adopting a food web approach permits testing whether predator-prey scaling among entire ecosystems (Fig. 1d), can be extrapolated from scaling relations among food web populations (Fig. 1c) - and vice versa - providing a basis to link biomass distributions across levels of biological organisation. We would expect that

population predator-prey scaling implies similar ecosystem predator-prey scaling in the limit of very simple systems, but that predictions are more involved in more complex food webs with variation in size structure, prey partitioning among predators and degree of omnivory.

In a simple, but not very realistic case, we consider the case of no omnivory, and no systematic change in size structure, where one predator population feeds exclusively on one prey population along a gradient. In this case, no matter how we aggregate prey is likewise how we aggregate predators, and so any aggregate predator-prey scaling will approach the same exponent as the largest population-level predator-prey exponent.

Assuming that all predator-prey populations scale with a similar exponent k , we can write aggregate prey ecosystem biomass (B), consisting of populations i , as $B = \sum_i B_i$. If size structure is invariant (e.g. a constant predator-prey body mass ratio), then the fraction of biomass in population i is independent of the total biomass density B : $f_i = B_i/B$.

It follows that population predator-prey scaling ($C_i = z_i B_i^k$, where C_i is predator and z_i is a constant coefficient for population i) implies community predator-prey scaling:

$$C = \sum_i C_i = \sum_i z_i B_i^k = \sum_i z_i (f_i B)^k = \left(\sum_i z_i (f_i)^k \right) B^k$$

The term in brackets on the right hand side should be a constant, so population scaling implies community scaling (at least in our highly simplified scenario).

It becomes more complicated when we consider that one predator population is sharing some or all prey populations with other predators. We have undertaken preliminary simulations where predator species share two or three prey species with other predators. Although we find that aggregate ecosystem predator-prey scaling still maintains the same scaling as the population level, we hesitate to speculate further given the many dimensions in which food web properties might vary. These include variation in omnivory and size structure, which we know are highly variable and possibly systematic. Further theoretical work is needed to know if constancy or lack of systematic variation in these properties is sufficient to predict ecosystem level scaling from the aggregate of population level scaling.

2) I don't think that the definitions of the different levels of analysis ("population level", "metapop" and so on) make much sense. I see what the authors are trying to do here, I just think the names are incorrect. At the population level, the data point is

actually a property of the predator-prey pair, not of any one population on its own. At the metapopulation level, there isn't an actual metapopulation component, at least as understood from classic metapopulation theory (collection of local populations connected by dispersal), nor would "metacommunity" be appropriate either since there is, as far as the authors can tell (and unless I'm missing something), actual dispersal between these food webs, or their data is blind to these effects. I think the only name I can get behind is the ecosystem one. All options I can think of are a bit verbose, but probably better than current choices: "intra food web", "inter food web" and "inter ecosystem" comparisons or something like that. NOTE: I retrospect, it was clear from continuing my read that the authors do have information on taxa that occur in multiple food webs. Whether these actually represent true meta-populations is a different issue, and would need to be better explained and discussed. Still, I argue that no relationship shown here occurs at the population level as for any slope to exist data is needed from more than one population, and regardless, any given predator-prey interaction is already occurring between at least two populations. Anyhow, this is just a wordsmithing issue.

Thank you for raising this. We have amended the terminology in line with these suggestions and the suggestion of Reviewer 1. We have removed 'metacommunity' when describing the intermediate scaling relation, which has subsequently been removed from the main text of the manuscript. We have also followed the suggestion here and use the terms 'population (intra-web) scaling' and 'ecosystem (inter-web) scaling' throughout.

See, for example, the text in the figure 1 legend:

(c) Population-level (intra-web) scaling relations are such that each data point represents the biomass of different predator taxa (nodes) plotted against the total biomass of their prey, within a single food web (see also Fig. 2). (d) Ecosystem-level (inter-web) predator-prey scaling relations represent the total biomass of all predators plotted against the total biomass of all prey for each distinct food web (see also Fig. 4).

4) I feel like a bit more handholding would be needed for the readers to understand where some of this variation may be coming from. For example, both freshwater and marine ecosystems show some negative slopes (smaller predator relative to prey!) but this doesn't show up in terrestrial systems. Yet, aquatic systems being gape limited, I would have hypothesized that negative slopes should not be found in such systems, but could occur in terrestrial systems (e.g., spiders eating birds, lions consuming elephants, while rare, it does occur). I suggest the authors 1) make some of the tested hypotheses more explicit so the results don't come out of the blue when presented, and at least discuss why some of the observed patterns make/doesn't make sense based on what we know about the way predator prey interactions occur in different ecosystems.

Thank you for this comment. Figure 2 displays the relationships between predator biomass and prey biomass. We believe the reviewer may have misunderstood this part of the analysis and is referring to predator and prey body mass relationships rather than our analysis, which concerns predator and prey biomass relations at a higher level of organisation. We acknowledge that there is notable variation in the exponent values within webs (see response to Reviewer 1 above) and indeed a small proportion of exponents in this analysis are negative. However, this does not mean that prey are larger than predators as suggested. Instead this indicates that predator biomass

decreases with prey biomass, which is plausible under some circumstances (e.g. where there is a mismatch between habitat availability and prey availability). We have now added clarification in several areas to avoid this common error.

Page 7, line 185:

"Specifically, we ask: (i) how does predator and prey biomass scale in complex food webs? (ii) what role does omnivory and variation in the size of predators relative to their prey have on the scaling pattern? (iii) does a general power-law hold across level of biological organization? and (iv) is the association between predator and prey biomass underpinned by changes in prey density or the average size of prey? Our results reveal fundamental similarities in the biomass structure of food webs in very different ecosystem types"

Page 8, line 209:

"To characterise population-level (intra-web) scaling (Fig. 1c), we constructed relationships between (\log_{10}) predator and (\log_{10}) prey biomass for each food web (Figs. S1 to S3) and fitted linear mixed-effects models to determine the scaling exponent (k) of the power-law for each ecosystem type in a single pass (Methods)."

MINOR

ABSTRACT

L-38: No need to state it's powerful. It can be powerful and it also has limitations. Thank you, we have made this change.

L-46: I think stating it's biomass matters. I suggest "more prey biomass supports proportionally less predator biomass", otherwise you need to say "fewer predators" instead of less, but then the analyses weren't done on abundances, which is why I suggest the phrasing above.

Thank you, we have made this change.

INTRO

l-64: Depends on how you define recent. 1yrs? 2yrs? 5yrs? Then no. Last decade? Then yes. But is within 10 years that recent? Definitely not dumping on Hatton et al, great paper and definite citation here. I'd just say a "previous" general finding since "recent" is on the eyes of the beholder.

We agree and have replaced "recent" with "previous"

DISCUSSION:

L-292: what do you mean flow is enhance to highly omnivory taxa? You mean "by" highly omnivory taxa? That omnivores enhance flow within food webs is trivially true and this analysis wasn't needed to say that given that omnivores prey upon multiple trophic levels, thus curt-circuiting intermediate TLs.

We have amended the text to now be specific about biomass stocks.

Revised text, page 14, line 343:

"Our results go beyond prior theoretical studies^{6,7} to provide empirical evidence that populations of highly omnivorous predators, as well as predator populations that feed

down the food web on smaller, more productive, prey (i.e. a high predator-to-prey body mass ratio) tend to attain higher biomass stocks”.

L-308-311: I think that going from your results to these implications is a major overreach.

We agree and have removed this from the discussion.

note: all Science papers in your reference list have (80-) right after the name if the journal. I get it, nobody cares, but I thought you should know your reference editor is doing this consistently.

Thank you for spotting this! We have now amended these and solved the issue.

Reviewer #3 (Remarks to the Author):

This is a high-quality manuscript that will be of considerable interest to a broad readership. The findings are novel, it is very well written and the analyses are robust. I have only 2 general points of (constructive) criticism:

Thank you for your very positive comments. We also believe the manuscript will be of appeal to a wide readership.

1) First, it is a pity that connections have not been drawn with theoretical and empirical work on size spectra given that much of this work has a similar focus to this manuscript and offers highly complementary insights;

We have now made connections between predator-prey biomass scaling and size spectra theory in the new Text Box. This includes a dedicated figure that gives an example of how sub-linear predator-prey biomass scaling and size spectra might be related.

Text box extract:

*“**Body size scaling theory** is focused on explaining the origin of size-scaling, such as the $\sim 3/4$ scaling of metabolism across classes of organisms^{18,19} or linking different variables through their scaling exponents to make novel predictions^{9,10}. Metabolic scaling theory would seem particularly relevant given the great variety of individual level rates that scale with body size near $3/4$, including fundamental variables such as growth and metabolism^{9,10}. However, these theories and empirical relations are focused at the individual level, and thus less relevant to aggregates of individuals in a community (but see^{17,20,21}. One exception is size-spectrum theory, which models the observation that biomass is approximately evenly distributed across logarithmic body size classes^{22,23}. This pattern has been observed for communities in water and on land²⁴ and appears to hold across all ocean life from bacteria to whales²⁵. Size spectrum theory often assumes that the predator-prey body mass ratio (PPmR), and trophic transfer efficiency (ratio of predator to prey production) is constant^{21,22}. These measures indicate from which size class energy is obtained, and how efficiently that energy is utilized by any given predator to maintain its biomass³. However, assuming both an even distribution of biomass across size classes, and a constant PPmR, suggests an unchanging trophic biomass pyramid (all else being equal), and is thus inconsistent with sub-linear predator-prey scaling. Such scaling means that the predator-prey biomass ratio (PPBR) is declining (Fig. Box 2 a), and therefore the shape of the biomass pyramid becomes more bottom heavy with enrichment (Fig. Box 2 e). To make size spectrum theory consistent with sublinear biomass scaling, we*

should vary the biomass spectrum slope (Fig. Box 2 f), mirroring changes in pyramid shape, or should change the variance of the PPmR (Fig. Box 2 g) across an enrichment gradient. However, there is, as yet, no evidence that size spectra slopes or PPmR vary systematically with enrichment.

Fig. Box 2 | Predator-prey scaling and the size-spectrum. Sublinear predator-prey scaling gives a declining predator-prey biomass ratio (PPBR slope < 0), as shown in (a), and a more bottom-heavy trophic biomass pyramid (b) across an enrichment gradient, as shown in (e). The biomass pyramid (b) can also be related to body size, since trophic interactions tend to be size structured (i.e. big eats small), which is especially the case in aquatic systems. Trophic level is thus related to body mass through the predator-prey body mass ratio, PPmR (c). To relate the body mass ratio, PPmR (c) to the biomass ratio, PPBR (a), we relate body mass to biomass through the biomass spectrum (d), which is the distribution of biomass across logarithmic body size classes. For PPBR to have a slope < 0 (a), either the biomass spectrum slope should change (f), or the variance of PPmR should change along an enrichment gradient (g). Alternatively, trophic transfer efficiency or other variables could change, which we have not considered.”

2) Second, the authors seem to implicitly assume that rock pools are broadly representative of marine ecosystems in general. This is not the case. Rock pools are essentially 2-dimensional and have strong system boundaries, and are more akin to streams and terrestrial foodwebs in these important respects. It is well-recognised that both dimensionality and 'open-ness' have important implications for food web structure and function and this needs to be recognised and considered in the discussion. It would be very interesting to also briefly hypothesise whether and how these relationships might change in other marine ecosystems.

Thank you for this helpful comment. In line with this comment (and comment from Reviewer 1), we have now added a paragraph about whether a similar pattern might hold in other marine systems and important differences to consider (dimensionality, openness and ontogenetic growth).

Page 14, line 350: “Further research would be instructive to understand if these are general patterns across different types of terrestrial and aquatic ecosystems. For instance, whilst rock pool food webs display very similar topological organization and PPmR scaling as open marine webs^{35,36}, we might expect different scaling patterns in pelagic marine webs where trophic interactions take place in three-dimensions³²; where ontogenetic diet shifts are common³⁷; and where food chains are long²⁶. Adapting our food-web approach to quantify biomass scaling among size

classes would likely be informative for tackling this”.

I believe that addressing these points will considerably improve what is already a good paper by broadening its relevance and better contextualizing it within a richer body of theoretical and empirical work.

In addition to these general points, I offer some specific suggestions:

- line 56: consider adding "and individuals" after "species" in recognition of the fact that in many marine taxa, trophic position is more strongly tied to individual body size (and life history stage) than to species. The importance of ontogenetic omnivory in indeterminate growers could be better considered throughout.

We have added species to this sentence. We have also discussed the importance of ontogenetic omnivory in the discussion (rather than introduction) to avoid disrupting the flow of the text (page 14, lines 350-360, see also revised text in response to comment above).

- line 78: as above, I suggest also mentioning ontogenetic omnivory here as this is a key feature of marine food webs.

As outlined in the response above, we have discussed the importance of ontogenetic omnivory in the discussion (rather than introduction) to avoid disrupting the flow of the text.

- lines 88-94: work on biomass spectra has been exploring these concepts for the better part of 50 years, and would warrant mention here.

We have now made specific mention to size spectra research here. New text, page 6, line 166: “*Size spectrum theory outlines a higher biomass can also potentially be sustained for large predators that feed ‘down’ the food web²⁸ - i.e. on the lowest trophic levels, which are typically the smallest size - because mass-specific production (and the production-to-biomass ratio) is greater for smaller organisms²⁹*”

- Figure 2-5 and methods: it wasn't entirely clear to me why the 3 ecosystem types were consistently modeled separately. Was there a strong reason for doing this instead of including ecosystem type as a fixed effect (with an interaction with prey biomass) in the models? This would give you an estimate of the overall cross-ecosystem slope (and arguably a more parsimonious way of doing this than by considering the average of the individual slopes) and also better enable comparing slopes and intercepts across ecosystems.

Thank you for this comment. We did in fact include ecosystem type as a fixed effect in the analysis, as suggested. We have modified the results and methods text to make this clearer. Revised text, page 8, line 209: “*To characterise population-level (intra-web) scaling (Fig. 1c), we constructed relationships between predator and prey biomass for each food web (Figs. S1 to S3) and fitted linear mixed-effects models to determine the scaling exponent (k) of the power-law for each ecosystem type in a single pass (Methods).*”

- line 239-240: I'm not sure that the statement regarding greater consistency in biomass structure than had been previously recognized is true for marine systems where consistency of biomass spectra has been appreciated for decades.

We have modified this sentence to highlight that we meant in fact “greater consistency across an enrichment gradient” that previously recognised (page 2, lines 66-67).

- Lines 261-263: could draw parallels with work examining 'escapes' from size spectra arising from subsidies here.

Thank you for this suggestion. We have now made reference to this work (Hocking et al. 2013) on page 16, line 379.

- Lines 311-313: I suggest tempering this final statement somewhat. Perhaps reword to "provides insight into baselines for structure...". I'm skeptical that the insights from rockpools shed light on baselines for marine ecosystems more broadly.

We have amended this sentence in line with this suggestion.

Page 15, line 361: “*The regularity in predator-prey scaling we observed could provide insight into baselines for the biomass structure of natural communities, which could be informative for assessing the effects of environmental impacts within ecological communities and ecological status*”

Papers that may be helpful to reference in addressing the above:

- Andersen KH, Berge T, Gonçalves RJ, Hartvig M, Heuschele J, Hylander S, Jacobsen NS, Lindemann C, Martens EA, Neuheimer AB, Olsson K, Palacz A, Prowse AEF, Sainmont J, Traving SJ, Visser AW, Wadhwa N & Kiørboe T (2016). Characteristic Sizes of Life in the Oceans, from Bacteria to Whales. *Annual Review of Marine Science* 8(1):217–241.

- Pawar S, Dell AI, & Van M. Savage (2012). Dimensionality of consumer search space drives trophic interaction strengths. *Nature* 486(7404):485–489.

- Jennings S, Mélin F, Blanchard JL, Forster RM, Dulvy NK & Wilson RW (2008). Global-scale predictions of community and ecosystem properties from simple ecological theory. *Proceedings of the Royal Society B: Biological Sciences* 275(1641):1375–1383.

- Polishchuk LV & Blanchard JL (2019). Uniting Discoveries of Abundance-Size Distributions from Soils and Seas. *Trends in Ecology & Evolution* 34(1):2–5.

We greatly appreciate these suggestions. We now include Pawar et al. (2012) and Polishchuk & Blanchard (2019), as well as additional papers on size spectra and marine systems, in the revised draft.

Reviewer comments, second round -

Reviewer #1 (Remarks to the Author):

(pls see attached file for an easier to read version)

The authors have adequately addressed many of the issues raised by me and the other reviewers. Their revised version now has a clearer logical structure, and includes a description of the theory that could and could not explain their observed pattern. The authors have also appropriately concentrated their analyses of scaling on predator-prey pairs within webs and on total predator to total prey biomasses in a web across different webs (removing the intermediate level). Further, the presentation of the results are now more nuanced, clearly specifying also the great variation around estimates. Similarly, the descriptions of the study systems used are more specific and correct, including a brief discussion on how they compare to other types of systems.

Personally, I would have preferred a study that first outlined theory and their predictions, and then tested these predictions in data (followed by a discussion of the findings). The authors do it the other way around; they present a pattern, and given that pattern, present which theory could have explained and which that could not have explained this pattern (in the text box). I leave it to the editor to decide whether this is a useful and clear way of presenting the work.

The structure of the results is now much improved, and mostly clearly presented. Reporting of the residual analyses, however, need to be corrected to not be misleading, in both the results, figure legends and the discussion (indicated in detailed comments, below).

Finally, I note that there is no discussion of the methods used, nor of how the conclusions may depend on these; e.g. how may the conclusions have been affected by the assumption made that interaction strengths were equal and each prey equally vulnerable to all its predators? The authors touch upon some of these, and make interesting additional analyses to address these, but that is completely omitted from the discussion. This is important to include, such that the readers can evaluate the robustness of the conclusions.

Detailed comments to authors:

Abstract: it is good that you now explain the underlying theory in the main text, however this doesn't come through in the abstract. I suggest you revise it slightly to clarify the theory-guided motivation of the study (e.g. on line 42, before 'we test'). One aspect of this is to explain what pattern would emerge from density-dependent growth, and if the aim of your study was to look for that particular pattern (to infer dd growth).

L51-52: the conclusion becomes a somewhat circular reasoning as it is so similar to the beginning of your abstract; suggest you end with a more specific conclusion to avoid that

L62-64: good that you end the paragraph specifying a knowledge gap. However, the previous part of the paragraph needs to help the reader understand why this is a gap, by reviewing available knowledge and earlier studies. Currently you don't do that, and the knowledge gap is therefore not convincing.

L70: suggest you remove the parentheses around 'double logarithmic scales' as you have very many parentheses in this single sentence

L73-74: this focuses on a pattern. To maintain the theory driven approach, and clarify it to the reader, you need to explain which processes that could lead up to that pattern. While referring to the new text box is good, you need to help the reader also in the main text, with a sentence or two explaining how your statement on L 74-76 would work, i.e. why does this pattern suggest a systematic form of density-dependent population growth?"

Text box: How can enrichment be represented by a change in the predator-prey interaction term – what's the rationale?

Textbox 1 ends with an interesting prediction, one which I guess you actually could test using your data; Does PPMR variance increase with enrichment?

L87-88: reference to the description of the methods used to generate the predictions in the text box is missing, suggest you insert that here (i.e. reference to Appendix S1)

L92-93: I presume you mean "...not consistent with sublinear predator-prey scaling...?"

L98: rephrase to "...equilibrium biomass scaling..."

L157: this paragraph 'However' should refer back to the previous paragraph in the main text (i.e. L65-77, and not the text box (as it may be layouted to be somewhere else on the page or even in an adjacent page). I presume the 'however' and reference to food chains is to relate it to early theoretical work. So was the earlier study by you, Hatton et al. Science, on food-chains? if so, then you could add that info on L67 to help the reader understand what it is you are now contrasting again, e.g L67: "Here, using a food chain model, predator biomass...ref8"

L162 replace with "... (Fig. 1b) and where energy..."

L185-L190: insert ; after the first two ? in the list of research questions, for readability

L196: "along a prey biomass gradient"

L212: a strong result of yours is that all the exponents are sub-linear; since you started out the study by pointing out that it was unknown whether this was the case, I suggest you start the results by presenting this finding. For example, rewrite to "Doing so revealed that the average exponents all were sublinear and that the exponent for each ecosystem type was similar, although..."

L227: to be consistent across figure legends, suggest you rephrase to "Population-level (intra-web) biomass scaling..."

L243-244: this is not strictly correct; your test & your illustration (Fig 3a-c) was of the *residuals* of predator biomass when explained by prey biomass, and how these residuals change with PPMR. So what this shows is that the (positive) deviations in predator biomass from the predicted relationship to prey biomass increased with PPMR. (You don't know whether predator biomass increases with only PPMR as explanatory factor, as you haven't done that analysis). So better rephrase this to e.g. "the predator biomass increased significantly more with prey biomass with increasing PPMR..." to report it accurately. Similarly

L247 needs to be corrected to "deviations in predator biomass from its scaling with prey biomass increasing with PPMR more strongly..." and on L249 to "...with deviations in predator biomass..."

L256: for understanding, please state explicitly "...population-level scaling of predator biomass with prey biomass (Table S5)."

L256 & L257: in both places correct to "...between residual predator biomass..."

L258: correct to "...biomass deviations increase significantly with PPMR..."

L264: could be good to remind the reader why you are looking for this pattern (the link to the theory)

L266: suggest you insert your finding on sublinear scaling here as well; i.e. "...revealed sublinear and remarkably similar..."

L277 & 285: suggest you in both places rewrite to "Ecosystem-level predator-prey biomass scaling..."

L306: I guess you mean "...for understanding of ..."

L308: correct to 'ecosystem type'

L338: specify as "...average prey body size..."

L347: please correct to "...biomass stocks than predicted by their prey biomass alone." (as this conclusion relies on your analyses of predator biomass *residuals*).

L348: similarly, this also needs to be corrected, e.g. to "...with predator biomass deviations increasing..."

L349: for better readability, suggest you rephrase slightly to "...in rock pool webs, whereas predator omnivory only proved to drive..."

L354-355 replace semicolons with commas, for readability

L417 & L418: also here suggest you aim to be consistent and rephrase to "Population-level (intra-web) biomass scaling..."

Supplement, Fig. S4 legend: suggest you use the sub-plot labels (a)-(d) in the corresponding places in the legend

Reviewer #2 (Remarks to the Author):

I've re-read with great interest the now greatly revised version of the paper by Perkins and collaborators where they show that predator-prey biomass scalings occur in both pairwise interactions (intra-web), and across food webs (inter-web). I believe this version to be much stronger, to get across the main points more clearly, and to also more clearly but more fairly put these interesting results in the context of the broader literature and existing theory. It is always immensely satisfying to work with responsive authors that take comments seriously and go beyond the call of duty, as was done here. Below I provide additional suggestions and thank the authors for the revisions.

MAJOR (as in needs to be explained or considered but aren't paper-breaking)

1) Box figure 2, I feel like some extra handholding is needed for readers (and this reviewer) to understand why increased variance in PPMR could lead to an increasingly bottom heavy pyramid. As I see it, the increased variance may not lead to a change in pyramid structure unless it's due to an increase in the skew of the PPMR, in which case it wouldn't just need a change in the second moment of the distribution, but a change in the third moment of the distribution as well. Does this make sense? In any case, this needs to be better explained.

2) As I stated before, I don't believe "population-level scaling" is a good description of what is being done in this paper: a scaling requires the interaction between at least two populations, and there is no such thing as a population of two interacting species. I propose that the "population" naming is dropped for good, in favor of something like "pairwise" scaling (each predator and each prey leads to a point).

3) the authors characterize the exponent of the relationship (intra-web) in freshwater, marine and terrestrial as being "similar". However, 0.75 (marine, terrestrial) is actually 25% larger than 0.66 (freshwater). Because the scalings are between 0 and 1, it seems like these are not that different, but this would be considered a large difference in any variable that isn't distributed in the [0,1] interval. Moreover, this, I believe, holds some information about the factors that determine these scalings across food webs and why they may differ, and I believe it would make sense to present it as this and discuss this result in its fullest extent. However, it is interesting that in the across-food web comparison these scalings do become similar and close to a 2/3 scaling.

MINOR:

I 39; needs comma after e.g.

I 67; why enrichment? Comes out of the blue here, perhaps ease reader a little here? Also, "Here" is a little equivocal. You clearly don't mean "in this paper" as you are talking about Hatton et al. Perhaps say something like "That previous work showed that..." NOTE: in continuing reading, I realized that what is missing is the fact that you mention enrichment repeatedly, but you never actually connect that enrichment with increasing prey biomass, which seems to be your point in box fig 2e. You should try to make this connection clearer in your intro, box, or both.

I 76; I suggest changing "and are intriguing" by "which is particularly interesting/intriguing (you choose) given the parallels..."

box fig 1; "A logistic carrying capacity" is unclear. You vary K, or you vary the functional form of logistic growth, but not "logistic K". Also, it seems like a and b are reverse (the labeling not the plots), as you state $K=1$ in a (but there is no K in model a, really), and you state K undefined for b (but there is a K in this model). Also, every K definition is messed up in the pdf for some reason? Just mentioning this so you check you .svg or source figure to make sure it's a conversion issue but not something that may end up in the published version.

I-134: declining "with biomass" correct? I think you need to say that, otherwise it's unclear.

box Fig 2: only fig 2a is called, but b, c and d aren't. However, you could add appropriate calls in lines 135 through 137 to do some extra handholding for your readers.

I-162: add "and" between comma and "where"

I 176: food web populations is a weird phrasing, it seems what you actually want to say is "among populations within food webs". This is related to some of the wording issues prevalent in the previous version of this MS.

I-190, 191: the last sentence of the introduction should better explain the unique, novel findings this paper has, which I don't believe are well captured by "Our results reveal fundamental similarities in the biomass structure of food webs in very different ecosystem types", which in and of itself would not be particularly novel.

Reviewer #3 (Remarks to the Author):

The manuscript has been substantially revised to address comments from myself and the other reviewers on the original submission. In particular, two new box/figure elements have been added that aim to link biomass scaling with simple predator-prey and size spectra models and theory, much of the text has been re-written. While I commend the authors for these efforts to address the reviewer comments and connect their work to a wider body of theoretical and empirical work, I unfortunately found much of the revised material confusing and difficult to follow (as I expand upon below).

Specific comments:

- I really struggled to understand both of the new boxes and their explanations in the text. Having pored over this, I do think these elements are potentially useful, but they need to be much better explained and (particularly in the case of box 1), better integrated with the rest of the manuscript.
- In the case of box 1, it is not immediately clear where the simulations referred to in the box subsequently in the text come from and how they relate to the rest of the methods as they are not currently mentioned in the main methods, nor is code provided for them with the scripts for the rest of the analyses in the supplementary materials. These simulations need to be better explained and integrated in to the manuscript. What are the predator and prey isoclines? How are they estimated? Why are they informative?

- For box 2 (and related text elsewhere), 2 key stumbling-blocks that I struggled with when trying to understand these parts of the manuscript were:

1. What the term 'enrichment' (and 'enrichment gradient') meant. This terminology wasn't used in the original submission and I found its use in this revision to be non-intuitive and ambiguous. If this terminology is retained, its intended meaning needs to be better explained/defined. Why is it used instead of 'higher prey biomass', 'higher primary production' or 'higher total biomass' (or a similar alternative that is less ambiguous)? One of the reasons I found this confusing is that, to me (as someone who works on food web models, including size-based models) it made me immediately think of energetic subsidies (i.e. production entering the food web other than via 'local' primary producers) which has important implications for size structure. I realise that this isn't the intended meaning (at least I don't think so), but this needs to be crystal clear.

2. The intended meaning when talking about change in size spectrum slope, PPMR, and biomass pyramids. I eventually figured out that (as far as I can tell) this is about how these parameters are expected to vary with total prey biomass. However, on first reading, this wasn't clear. This was dissonant because these are static scaling models of size spectra.

- specific suggested changes for box 2 fig:

- replace 'enrichment' terminology with something more self explanatory and directly comparable to the axis labels in the figure elements.

- add descriptive 'column headings' for the 2 sides of the figure (1 for panels a to d; 1 for panels e to g). For panels e to g this should clarify what the changes in pyramid shape, spectrum slope and PPMR in the subsequent sub-headings is with respect to.

- Reword second last sentences of the caption to "For PPMR to have a slope <0 , with increasing prey biomass: (a), either ...". Having the mention of enrichment gradient at the end of the sentences as currently written makes it unclear whether this is for just PPMR or slope too.

- Reword last sentence to specify that this is "with increasing total prey biomass"

- As one of the other reviewers pointed out, there is some similarity between this manuscript and Hatton et al 2015. Using the title of the Hatton et al 2015 manuscript as the running head for this manuscript doesn't help to that end.

- line 125: reword "which models the observation that biomass..." with "which aims to explain the observation that, for whole ecosystems, biomass...". This is to address two points: (1) the theory is the basis for developing models, but does not directly model observations; and (2) important exceptions to this pattern are evident in some assemblages/communities.

- line 127: replace "communities" with "ecosystems". As above, this is because the pattern of approximately equal biomass is at the ecosystem scale; communities and assemblages within ecosystems may show very different patterns, particularly if there are energetic subsidies.

- Line 127: Blanchard et al's 2017 TREE article (which I pointed the authors toward in my previous review) should be cited here as it presents a comprehensive synthesis of the branches of size spectrum theory and models, as well as a 'taxonomy' for these models. The branch of size spectrum theory and models being described here is for 'static' models (other branches have quite different assumptions regarding PPMR and transfer efficiency). Polischuk & Blanchard 2019 (which is currently cited here) was a commentary on the 2017 article.

Blanchard JL, Heneghan RF, Everett JD, Trebilco R, Richardson AJ (2017) From Bacteria to Whales: Using Functional Size Spectra to Model Marine Ecosystems. *Trends in Ecology & Evolution* 32:174–186.

- Line 127-128: as noted above, I suggest rewording from "Size spectrum theory often assumes" to "Static scaling models of size spectra assume that..."

- Line 129: suggest rewording from "is constant" to "... do not vary with total system biomass" to make it clearer what you're suggesting PPMR and TE are constant with respect to. Note also that the paper by Barnes et al. that is cited elsewhere suggests that rather than being constant, a compensatory relationship between TE and PPMR could explain observed consistency in size spectrum slopes

- line 139-140: this is a point where it is really important to clarify how enrichment is (or isn't) related to subsidies. From a size spectrum perspective, an energetic subsidy (i.e. production entering the size spectrum at a size class above primary producers) would be expected to affect size spectrum slopes.

- Line 139-140: note also that there smaller mean predator-prey body size ratios are characteristic of more stable environments

Jennings S, Warr KJ (2003) Smaller predator-prey body size ratios in longer food chains. *Proc R*

Soc Lond B 270:1413–1417. There's

- line 167: note that Trebilco et al 2013 (which you cite elsewhere) articulated this hypothesis prior to Woodson et al.

We have made the following changes:

- **Revised the terminology for the scaling relations, removing the terms 'population-level' and 'ecosystem' level.**
 - **Removed the Text Box and distributed the text among introduction and discussion sections as well as a dedicated appendix in the Supplementary Material.**
 - **Made minor changes to the manuscript format so that it complies with the required format for Nature Communications as detailed in the checklist.**
-

We provide a point-by-point response to reviewers' comments below.

REVIEWER COMMENTS

Reviewer #1 (Remarks to the Author):

(pls see attached file for an easier to read version)

The authors have adequately addressed many of the issues raised by me and the other reviewers. Their revised version now has a clearer logical structure, and includes a description of the theory that could and could not explain their observed pattern. The authors have also appropriately concentrated their analyses of scaling on predator-prey pairs within webs and on total predator to total prey biomasses in a web across different webs (removing the intermediate level). Further, the presentation of the results are now more nuanced, clearly specifying also the great variation around estimates. Similarly, the descriptions of the study systems used are more specific and correct, including a brief discussion on how they compare to other types of systems.

Thank you for this comment.

Personally, I would have preferred a study that first outlined theory and their predictions, and then tested these predictions in data (followed by a discussion of the findings). The authors do it the other way around; they present a pattern, and given that pattern, present which theory could have explained and which that could not have explained this pattern (in the text box). I leave it to the editor to decide whether this is a useful and clear way of presenting the work.

Thank you for this comment, we have not changed the order in our rationale but hope the corrections detailed below result in a better flow overall.

The structure of the results is now much improved, and mostly clearly presented. Reporting of the residual analyses, however, need to be corrected to not be misleading, in both the results, figure legends and the discussion (indicated in detailed comments, below).

Thank you for this comment, we explain how we addressed this in the detailed comments below.

Finally, I note that there is no discussion of the methods used, nor of how the conclusions may depend on these; e.g. how may the conclusions have been affected by the assumption made that interaction strengths were equal and each prey equally vulnerable to all its predators? The authors touch upon some of these, and make interesting additional analyses to address these, but that is completely omitted from the discussion. This is important to include, such that the readers can evaluate the robustness of the conclusions.

We now provide a new paragraph in the discussion addressing how the conclusions of the study might be affected by some of the assumptions made, which were necessary given the data currently available.

New paragraph, page 13, line 666:

“Our study, which takes a first step towards investigating predator-prey biomass scaling in complex food webs, has some notable limitations. First, information on the weighting of feeding links was not available for the food webs studied here, and a more comprehensive investigation should require specific interactions strengths and vulnerabilities of each species, data that is, as yet, unavailable. Although we have examined alternative assumptions in how these factors are treated, with no loss in generality (Table S5), it is possible that these variables could play an important role. Second, information on the biomass of all basal resources was also not generally available, so our analysis focused on higher trophic predators feeding on (animal) prey. While our approach may equally apply more generally to consumers and resources (e.g. aquatic snails feeding on biofilm), further work is required to test the generality of the empirical patterns we observed using more detailed datasets where this information, and data on interaction strengths, is widely available.”

Detailed comments to authors:

Abstract: it is good that you now explain the underlying theory in the main text, however this doesn't come through in the abstract. I suggest you revise it slightly to clarify the theory-guided motivation of the study (e.g. on line 42, before 'we test'). One aspect of this is to explain what pattern would emerge from density-dependent growth, and if the aim of your study was to look for that particular pattern (to infer dd growth).

Thank you for this comment. We have revised the abstract to outline the underlying theory / expectations. Revised extract, page 1, line 43:

“We test whether sub-linear scaling between predator and prey biomass (a potential signal of density-dependent processes) emerges within ecosystem types and across levels of biological organisation”.

L51-52: the conclusion becomes a somewhat circular reasoning as it is so similar to the beginning of your abstract; suggest you end with a more specific conclusion to avoid that

We have revised the final sentence of the abstract to avoid circular reasoning. Page 1, line 50: “We test whether sub-linear scaling between predator and prey biomass (a potential signal of density-dependent processes) emerges within ecosystem types and across levels of biological organisation”

L62-64: good that you end the paragraph specifying a knowledge gap. However, the previous part of the paragraph needs to help the reader understand why this is a gap, by reviewing available knowledge and earlier studies. Currently you don't do that, and the knowledge gap is therefore not convincing.

Thank you. We have now revised the final sentence of that paragraph (page 2, line 79) to make it clearer why there is a gap in our knowledge: “However, the principal mechanisms responsible for driving these patterns in natural systems remains uncertain because of a lack of empirical data, and investigations of how these patterns may change along environmental gradients are still in their infancy.”

L70: suggest you remove the parentheses around ‘double logarithmic scales’ as you have very many parentheses in this single sentence

Removed. Thank you.

L73-74: this focuses on a pattern. To maintain the theory driven approach, and clarify it to the reader, you need to explain which processes that could lead up to that pattern. While referring to the new text box is good, you need to help the reader also in the main text, with a sentence or two explaining how your statement on L 74-76 would work, i.e. why does this pattern suggest a systematic form of density-dependent population growth”?

In line with the Editor’s request, we have now removed the text box. We now provide a full explanation of the processes that can give rise to the sub-linear scaling pattern.

Page 2, line 91: “These empirical patterns could be underpinned by systematic changes in total prey production available to predators^{8,9} – that is, because predator biomass and prey productivity are linearly related^{8,10}; if predator biomass is sub-linearly related to prey biomass (Fig. 1a), then prey productivity should also be sub-linearly related to prey biomass. Two possible reasons explain sub-linear scaling of prey productivity with prey biomass: (1) constraints on individual rates which scale allometrically with body mass¹¹ and could lead to sub-linear community scaling with a systematic relation of body size to biomass¹², and/or (2) density dependent effects - i.e. submaximal individual growth⁸. Hatton et al.⁸ found that mean prey body mass varied little over the prey biomass gradient and thus their results are consistent with the role of density dependent processes driving sub-linear predator-prey biomass scaling.”

Text box: How can enrichment be represented by a change in the predator-prey interaction term – what’s the rationale?

We have revised this text, which now occurs in a dedicated appendix (Appendix S2) in the supplementary material: “For a given model (Appendix Table 1), we relate the equilibrium predator and prey biomass across a simulated gradient of total community biomass, and ask whether it recovers the empirical scaling observed. A community biomass or ‘enrichment’ gradient can be modeled by varying a parameter in the prey growth term (Appendix Table 1), which may be exponential or logistic, or else in the predator-prey interaction term, which may be a linear or saturating functional response 1–3.”

Textbox 1 ends with an interesting prediction, one which I guess you actually could test using your data; Does PPMR variance increase with enrichment?

We agree this is an interesting prediction, but to not overcomplicate the overall story, we refrain from testing this here as it is not the main focus of the study.

L87-88: reference to the description of the methods used to generate the predictions in the text box is missing, suggest you insert that here (i.e. reference to Appendix S1).

This text now occurs in a dedicated appendix (Appendix S2). Within this, we have added reference to Appendix Table 1, which includes the description of the methods used to generate the predictions.

L92-93: I presume you mean "...not consistent with sublinear predator-prey scaling..."?
Changed to "...not consistent with sublinear predator-prey scaling..."

L98: rephrase to "...equilibrium biomass scaling..."
Changed

L157: this paragraph 'However' should refer back to the previous paragraph in the main text (i.e. L65-77, and not the text box (as it may be layouted to be somewhere else on the page or even in an adjacent page). I presume the 'however' and reference to food chains is to relate it to early theoretical work. So was the earlier study by you, Hatton et al. Science, on food-chains? if so, then you could add that info on L67 to help the reader understand what it is you are now contrasting again, e.g L67: "Here, using a food chain model, predator biomass...ref8"

The text box has been removed, so it is now clearer that 'however' refers to the previous paragraph.

L162 replace with "... (Fig. 1b) and where energy..."
Replaced. Page 4, line 207.

L185-L190: insert ; after the first two ? in the list of research questions, for readability

We have now inserted ';' between questions (page 3, line 228):

"Specifically, we ask: (i) how does predator and prey biomass scale within (Fig. 1c) and across (Fig. 1d) complex food webs? and does a general power-law hold across levels of biological organization; (ii) what role does omnivory and predator-prey body mass ratios have on the scaling pattern? and (iii) is the association between predator and prey biomass underpinned by changes in prey density or the average size of prey?"

L196: "along a prey biomass gradient"

Changed: page 4, line 263.

L212: a strong result of yours is that all the exponents are sub-linear; since you started out the study by pointing out that it was unknown whether this was the case, I suggest you start the results by presenting this finding. For example, rewrite to "Doing so revealed that the average exponents all were sublinear and that the exponent for each ecosystem type was similar, although..."

Changed: page 5, line 313.

L227: to be consistent across figure legends, suggest you rephrase to "Population-level (intra-web) biomass scaling..."

Changed: page 6, line 340.

L243-244: this is not strictly correct; your test & your illustration (Fig 3a-c) was of the *residuals* of predator biomass when explained by prey biomass, and how these residuals change with PPMR. So what this shows is that the (positive) deviations in predator biomass from the predicted relationship to prey biomass increased with PPMR. (You don't know whether predator biomass increases with only PPMR as explanatory factor, as you haven't

done that analysis). So better rephrase this to e.g. “the predator biomass increased significantly more with prey biomass with increasing PPMR...” to report it accurately. Similarly L247 needs to be corrected to “deviations in predator biomass from its scaling with prey biomass increasing with PPMR more strongly...” and on L249 to “...with deviations in predator biomass...”

Thank you for this comment and we agree. We have now rephrased these sentences to make it clear that the analysis involves predator biomass residuals. Page 6, lines 356 & 370.

L256: for understanding, please state explicitly “...population-level scaling of predator biomass with prey biomass (Table S5).”

Changed: page 7, line 377.

L256 & L257: in both places correct to “...between residual predator biomass...”

Corrected: page 6, line 378.

L258: correct to “...biomass deviations increase significantly with PPMR...”

Corrected: page 6, line 380.

L264: could be good to remind the reader why you are looking for this pattern (the link to the theory).

Sentence amended to include this. Page 8, line 395: “*To investigate if similar predator-prey biomass scaling patterns emerge across webs, and thus whether within-web patterns can be scaled-up to whole ecosystems (Appendix S1), we summed the biomass of all predators and all prey within each food web.*”

L266: suggest you insert your finding on sublinear scaling here as well; i.e. “...revealed sublinear and remarkably similar...”

Changed: page 8, line 398.

L277 & 285: suggest you in both places rewrite to “Ecosystem-level predator-prey biomass scaling...”

This has now been changed to ‘across-web predator-prey biomass scaling’, in line with the Editor’s request: page 8, line 410.

L306: I guess you mean “...for understanding of ...”

Changed: page 10, line 462.

L308: correct to ‘ecosystem type’

Changed: page 10, line 462.

L338: specify as “...average prey body size...”

Changed: page 12, line 610.

L347: please correct to “...biomass stocks than predicted by their prey biomass alone.” (as this conclusion relies on your analyses of predator biomass *residuals*).

Changed: page 12 line 620.

L348: similarly, this also needs to be corrected, e.g. to “...with predator biomass deviations increasing...”

Changed: page 12, line 621.

L349: for better readability, suggest you rephrase slightly to "...in rock pool webs, whereas predator omnivory only proved to drive..."

Changed: page 12, line 622.

L354-355 replace semicolons with commas, for readability

Change made

L417 & L418: also here suggest you aim to be consistent and rephrase to "Population-level (intra-web) biomass scaling..."

This has now been changed to 'within-web predator-prey biomass scaling', in line with the Editor's request: page 15, line 4731.

Supplement, Fig. S4 legend: suggest you use the sub-plot labels (a)-(d) in the corresponding places in the legend.

Sub-plot labels now added to Fig. S4 legend.

Reviewer #2 (Remarks to the Author):

I've re-read with great interest the now greatly revised version of the paper by Perkins and collaborators where they show that predator-prey biomass scalings occur in both pairwise interactions (intra-web), and across food webs (inter-web). I believe this version to be much stronger, to get across the main points more clearly, and to also more clearly but more fairly put these interesting results in the context of the broader literature and existing theory. It is always immensely satisfying to work with responsive authors that take comments seriously and go beyond the call of duty, as was done here. Below I provide additional suggestions and thank the authors for the revisions.

We would like to thank the reviewer for their assessment of the paper and once again for comments which have helped improve the paper.

MAJOR (as in needs to be explained or considered but aren't paper-breaking)

1) Box figure 2, I feel like some extra handholding is needed for readers (and this reviewer) to understand why increased variance in PPMR could lead to an increasingly bottom heavy pyramid. As I see it, the increased variance may not lead to a change in pyramid structure unless it's due to an increase in the skew of the PPMR, in which case it wouldn't just need a change in the second moment of the distribution, but a change in the third moment of the distribution as well. Does this make sense? In any case, this needs to be better explained.

Thank you for raising this point. We have revised the text describing the patterns shown in this figure (now Appendix 2, Fig. 2) to make it clearer how changes in variance of PPMR (or more specifically variance in the size of prey consumed by predator) can give rise to a change in pyramid shape.

Appendix 2 revised text: "If the prey base of any given predator is constant, then the slope of the biomass pyramid should become more negative (Appendix Fig. 2 f), paralleling changes in pyramid shape (Appendix Fig. 2 e). If instead the biomass spectrum slope is constant, then a higher biomass indicates predators should be obtaining prey from a larger set of size classes to maintain a more bottom-heavy biomass pyramid. This translates into a decline in variance of PPMR at higher biomass (Appendix Fig. 2 g). However, there is, as yet no evidence that size spectra slopes or PPMR vary systematically with total community biomass.

More likely than the possible variations in these simple static properties (Appendix Fig. 2), are changes in dynamical variables along a biomass gradient, such as flux rates, trophic transfer efficiencies and/or productivities with biomass."

We have also added the following sentence to the figure caption: "***In (c), we consider the variation in prey body mass relative to two different predator masses, with the larger predator having a much broader prey base.***"

2) As I stated before, I don't believe "population-level scaling" is a good description of what is being done in this paper: a scaling requires the interaction between at least two populations, and there is no such thing as a population of two interacting species. I propose that the "population" naming is dropped for good, in favor of something like "pairwise" scaling (each predator and each prey leads to a point).

Thank you. We have now changed the terminology (in line also with the Editor's comment) and use 'within-web scaling' and 'across-web scaling' to make it clearer how the different relations are constructed.

3) the authors characterize the exponent of the relationship (intra-web) in freshwater, marine and terrestrial as being "similar". However, 0.75 (marine, terrestrial) is actually 25% larger than 0.66 (freshwater). Because the scalings are between 0 and 1, it seems like these are not that different, but this would be considered a large difference in any variable that isn't distributed in the [0, 1] interval. Moreover, this, I believe, holds some information about the factors that determine these scalings across food webs and why they may differ, and I believe it would make sense to present it as this and discuss this result in its fullest extent. However, it is interesting that in the across-food web comparison these scalings do become similar and close to a 2/3 scaling.

We agree that it is intriguing that the scaling exponents become lower for the across-web scaling patterns. However, statistically speaking, the average exponents do not differ between these scaling relations or between ecosystem types. Consequently, we refrain from adding additional text on this to the discussion to avoid reporting a mixed message.

MINOR:

l 39; needs comma after e.g.
added

l 67; why enrichment? Comes out of the blue here, perhaps ease reader a little here? Also, "Here" is a little equivocal. You clearly don't mean "in this paper" as you are taking about Hatton et al. Perhaps say something like "That previous work showed that..." NOTE: in continuing reading, I realized that what is missing is the fact that you mention enrichment repeatedly, but you never actually connect that enrichment with increasing prey biomass, which seems to be your point in box fig 2e. You should try to make this connection clearer in your intro, box, or both.

We have removed 'here' from this sentence and cite the previous paper (Hatton et al.) to avoid confusion. We have removed the term 'enrichment' from the manuscript and instead explicitly refer to 'increasing prey biomass' throughout.

Page 2, line 83: "A previous finding highlights remarkable regularity in how the ratio of predator-to-prey biomass changes across a gradient of prey biomass in both aquatic and terrestrial systems."

l 76; I suggest changing "and are intriguing" by "which is particularly interesting/intriguing (you choose) given the parallels..."

Changed to 'which is particularly intriguing given the parallels...'

box fig 1; "A logistic carrying capacity" is unclear. You vary K, or you vary the functional form of logistic growth, but not "logistic K". Also, it seems like a and b are reverse (the labeling not the plots), as you state $K=1$ in a (but there is no K in model a, really), and you state K undefined for b (but there is a K in this model). Also, every K definition is messed up in the pdf for some reason? Just mentioning this so you check you .svg or source figure to make sure it's a conversion issue but not something that may end up in the published version.

Thank you, we have amended the source figure and now there shouldn't be a conversion issue. This figure (now Appendix 2, Fig. 1) refers to k , the exponent of the predator-prey power law exponent and not K (carrying capacity parameter in growth models), which we believe the reviewer is referring to. We have amended the legend to make this clearer: "*Appendix Fig. 1 | Simple predator-prey model predictions across simulated biomass gradients. Gradients are simulated by varying interaction strength in (a) and (d), and by varying the carrying capacity for prey logistic growth in (b) and (c). Models (a) to (c) do not make predictions consistent with sublinear predator-prey biomass scaling (predator-prey biomass scaling exponent, k , either equal to 1 or undefined). Model (d) makes realistic predictions ($k < 1$) but requires that prey growth is sub-exponential, which is an assumption that requires new theory. Predator-prey model equations used to generate the phase space are provided in Appendix Table 1.*"

I-134: declining "with biomass" correct? I think you need to say that, otherwise it's unclear.
Correct, and added (Appendix S2)

box Fig 2: only fig 2a is called, but b, c and d aren't. However, you could add appropriate calls in lines 135 through 137 to do some extra handholding for your readers.

Thank you for this suggestion. We have revised the text to help the reader navigate the concepts shown in the figure. Appendix 2:

"Such scaling means that the predator-prey biomass ratio (PPBR; not to be confused with PPMR) is declining with prey biomass (Appendix Fig. 2 a), and therefore the shape of the biomass pyramid (Appendix Fig. 2 b) becomes more bottom heavy with enrichment (Appendix Fig. 2 e). To make size spectrum theory consistent with sub-linear biomass scaling, the biomass spectrum slope (Appendix Fig. 2 d) should vary (Appendix Fig. 2 f), mirroring changes in pyramid shape (Appendix Fig. 2 e), or the variance of the PPMR should change (Appendix Fig. 2 g) across a community biomass enrichment gradient."

I-162: add "and" between comma and "where".

Corrected (pointed out by Review 1 also)

I 176: food web populations is a weird phrasing, it seems what you actually want to say is "among populations within food webs". This is related to some of the wording issues prevalent in the previous version of this MS.

Agree, and this sentence has been removed in revising the introduction.

I-190, 191: the last sentence of the introduction should better explain the unique, novel findings this paper has, which I don't believe are well captured by "Our results reveal fundamental similarities in the biomass structure of food webs in very different ecosystem types", which in and of itself would not be particularly novel.

We have revised the last sentence of the introduction to focus more clearly on the novel finding of our study that similar scaling patterns emerge across levels of organisation. Page 4, line 257: "*Our results reveal fundamental similarities in predator and prey biomass scaling within and across diverse food webs, providing a basis to link biomass distributions across levels of biological organization (Appendix S1).*"

Reviewer #3 (Remarks to the Author):

The manuscript has been substantially revised to address comments from myself and the other reviewers on the original submission. In particular, two new box/figure elements have been added that aim to link biomass scaling with simple predator-prey and size spectra models and theory, much of the text has been re-written. While I commend the authors for these efforts to address the reviewer comments and connect their work to a wider body of theoretical and empirical work, I unfortunately found much of the revised material confusing and difficult to follow (as I expand upon below).

We have revised the manuscript following the reviewers comments and are confident that the new version has a clear rationale.

Specific comments:

- I really struggled to understand both of the new boxes and their explanations in the text. Having pored over this, I do think these elements are potentially useful, but they need to be much better explained and (particularly in the case of box 1), better integrated with the rest of the manuscript.

We have now integrated some of the text into the main manuscript and added a dedicated appendix (Appendix S2) where we expand on the concepts and theory.

- In the case of box 1, it is not immediately clear where the simulations referred to in the box subsequently in the text come from and how they relate to the rest of the methods as they are not currently mentioned in the main methods, nor is code provided for them with the scripts for the rest of the analyses in the supplementary materials. These simulations need to be better explained and integrated in to the manuscript. What are the predator and prey isoclines? How are they estimated? Why are they informative?

Thank you for raising this. The dedicated appendix covering this topic (Appendix S2) now contains both a visual representation of the simulations (Appendix Fig. 1) as well as the models underlying these (Appendix Table 1). The latter includes how the prey isoclines are estimated and how they link to predator-prey biomass scaling:

***“Appendix Table 1 | Predator-prey model equations used to generate the phase space isoclines in Appendix Fig. 1. In addition to the classic Lotka-Volterra model^{4,5} (a), we include a typical top-down control model such as Rosenzweig-MacArthur^{1,3} (b), and a bottom-up control model such as ratio-dependence^{6,7} (c). Finally, we show a modified Lotka-Volterra model, where prey growth is sublinear ($k < 1$)². In all equations, B is prey biomass and C is predator biomass. In addition, r is a prey growth constant (with different units in a-d); q is predator-prey interaction strength; g is predator growth conversion efficiency; h is sometimes referred to as predator handling time, controlling the saturation of the functional response; and m is predator mortality rate. A biomass gradient is modeled by varying q in (a) and (d), and by varying K in (b) and (c).*”**

Appendix Fig. 1 | Simple predator-prey model predictions across simulated biomass gradients. Gradients are simulated by varying interaction strength in (a) and (d), and by varying the carrying capacity for prey logistic growth in (b) and (c). Models (a) to (c) do not make predictions consistent with sublinear predator-prey biomass scaling (predator-prey biomass scaling exponent, k , either equal to 1 or undefined). Model (d) makes realistic predictions ($k < 1$) but requires that prey growth is sub-exponential, which is an assumption that requires new theory. Predator-prey model equations used to generate the phase space are provided in Appendix Table 1.”

- For box 2 (and related text elsewhere), 2 key stumbling-blocks that I struggled with when

trying to understand these parts of the manuscript were:

1. What the term 'enrichment' (and 'enrichment gradient') meant. This terminology wasn't used in the original submission and I found its use in this revision to be non-intuitive and ambiguous. If this terminology is retained, its intended meaning needs to be better explained/defined. Why is it used instead of 'higher prey biomass', 'higher primary production' or 'higher total biomass' (or a similar alternative that is less ambiguous)? One of the reasons I found this confusing is that, to me (as someone who works on food web models, including size-based models) it made me immediately think of energetic subsidies (i.e. production entering the food web other than via 'local' primary producers) which has important implications for size structure. I realise that this isn't the intended meaning (at least I don't think so), but this needs to be crystal clear.

We agree that the use of the term 'enrichment gradient' was a little ambiguous in the previous version. We now use the term 'higher prey biomass' or 'higher community biomass' in most instances and only refer to an enrichment gradient when it is clearly defined.

Revised text (Appendix S2): "Predator-prey theory models the dynamics of feeding interactions, and has traditionally focused on two distinct trophic levels, rather than on networks of highly omnivorous food webs 1. For a given model (Appendix Table 1), we relate the equilibrium predator and prey biomass across a simulated gradient of total community biomass, and ask whether it recovers the empirical scaling observed. A community biomass or 'enrichment' gradient can be modeled by varying a parameter in the prey growth term (Appendix Table 1), which may be exponential or logistic, or else in the predator-prey interaction term, which may be a linear or saturating functional response."

2. The intended meaning when talking about change in size spectrum slope, PPMR, and biomass pyramids. I eventually figured out that (as far as I can tell) this is about how these parameters are expected to vary with total prey biomass. However, on first reading, this wasn't clear. This was dissonant because these are static scaling models of size spectra.

- specific suggested changes for box 2 fig:

- replace 'enrichment' terminology with something more self explanatory and directly comparable to the axis labels in the figure elements.

- add descriptive 'column headings' for the 2 sides of the figure (1 for panels a to d; 1 for panels e to g). For panels e to g this should clarify what the changes in pyramid shape, spectrum slope and PPMR in the subsequent sub-headings is with respect to.

- Reword second last sentences of the caption to "For PPMR to have a slope <0 , with increasing prey biomass: (a), either ...". Having the mention of enrichment gradient at the end of the sentences as currently written makes it unclear whether this is for just PPMR or slope too.

Thank you for these suggestions. We have taken these onboard and revised the figure and figure caption:

“Appendix Fig. 2 | Predator-prey scaling and the size-spectrum. Sub-linear predator-prey scaling gives a declining predator-prey biomass ratio (a) along a biomass gradient ($PPBR$ slope < 0). This biomass scaling implies the trophic biomass pyramid (b) becomes more bottom-heavy with increasing biomass, as shown in (e). The biomass pyramid (b) can also be related to body size, since in aquatic systems, big eats small. Trophic level is thus broadly related to body mass through the predator-prey body mass ratio, $PPmR$ (c). In (c), we consider the variation in prey body mass relative to two different predator masses, with the larger predator having a much broader prey base. To relate the variation in prey body mass for any given predator (c) to the biomass ratio, $PPBR$ (a), we relate body mass to biomass through the biomass spectrum (d), which is the distribution of biomass across logarithmic body size classes, known to be approximately even. For $PPBR$ to have slope < 0 (sublinear predator-prey scaling, as in a), there are at least two possible corollaries. If the variation in prey body mass is constant, then it implies that the slope of the biomass spectrum should become more negative, mirroring changes in pyramid shape (f). If on the other hand, the biomass spectrum slope is constant, then the variation in prey body mass for any given predator should increase with increasing biomass, translating into a smaller $PPmR$ variation (g). This means that at high biomass, any given predator obtains prey from more size classes than at low biomass, and so the predator-prey biomass pyramid becomes more bottom-heavy. Axes are \log in (a), (c) and (d).”

- Reword last sentence to specify that this is "with increasing total prey biomass"
Complete: see Appendix 2, Fig. 1 caption (also above)

- As one of the other reviewers pointed out, there is some similarity between this manuscript and Hatton et al 2015. Using the title of the Hatton et al 2015 manuscript as the running head for this manuscript doesn't help to that end.

Running title changed to “predator-prey biomass scaling”

- line 125: reword "which models the observation that biomass..." with "which aims to explain the observation that, for whole ecosystems, biomass...". This is to address two points: (1) the theory is the basis for developing models, but does not directly model observations; and (2) important exceptions to this pattern are evident in some assemblages/communities.

Change made (see Appendix S2).

- line 127: replace "communities" with "ecosystems". As above, this is because the pattern of approximately equal biomass is at the ecosystem scale; communities and assemblages within ecosystems may show very different patterns, particularly if there are energetic subsidies.

We agree, thanks for spotting this. Community has been changed to ecosystems.

- Line 127: Blanchard et al's 2017 TREE article (which I pointed the authors toward in my previous review) should be cited here as it presents a comprehensive synthesis of the branches of size spectrum theory and models, as well as a 'taxonomy' for these models. The branch of size spectrum theory and models being described here is for 'static' models (other branches have quite different assumptions regarding PPMR and transfer efficiency). Polischuk & Blanchard 2019 (which is currently cited here) was a commentary on the 2017 article.

Blanchard JL, Heneghan RF, Everett JD, Trebilco R, Richardson AJ (2017) From Bacteria to Whales: Using Functional Size Spectra to Model Marine Ecosystems. *Trends in Ecology & Evolution* 32:174–186.

Thank you for clarifying this. Blanchard et al (2017) is now cited here (Appendix S2).

- Line 127-128: as noted above, I suggest rewording from "Size spectrum theory often assumes" to "Static scaling models of size spectra assume that..."

Changed made (page 11, line 540)

- Line 129: suggest rewording from "is constant" to "... do not vary with total system biomass" to make it clearer what you're suggesting PPMR and TE are constant with respect to. Note also that the paper by Barnes et al. that is cited elsewhere suggests that rather than being constant, a compensatory relationship between TE and PPMR could explain observed consistency in size spectrum slopes.

Changed (page 11, line 543)

- line 139-140: this is a point where it is really important to clarify how enrichment is (or isn't) related to subsidies. From a size spectrum perspective, an energetic subsidy (i.e. production entering the size spectrum at a size class above primary producers) would be expected to affect size spectrum slopes.

We now avoid using the term enrichment in the manuscript and only use it in the Appendix S2 when it is explicitly defined.

- Line 139-140: note also that there smaller mean predator–prey body size ratios are characteristic of more stable environments

Jennings S, Warr KJ (2003) Smaller predator-prey body size ratios in longer food chains. *Proc R Soc Lond B* 270:1413–1417.

Thank you for this suggestion.

- line 167: note that Trebilco et al 2013 (which you cite elsewhere) articulated this hypothesis prior to Woodson et al.

Trebilco et al 2013 is now added here (page 3, line 215)

Reviewer comments, third round -

Reviewer #3 (Remarks to the Author):

I commend the authors on their thorough and considered efforts to revise the manuscript to address all comments raised by myself and the other reviewers on the previously revised version of this paper. This updated version was a pleasure to read and I am sure it will be of considerable interest to the Nature Communications readership. I have no further concerns and look forward to seeing this work published.